# MicroRNA-mediated control of developmental lymphangiogenesis

**Hyun Min Jung, Ciara T Hu, Alexandra M Fister, Andrew E Davis, Daniel Castranova, Van N Pham, Lisa M Price, Brant M Weinstein\***

Division of Developmental Biology, Eunice Kennedy Shriver National Institute of Child Health and Human Development, National Institutes of Health, Bethesda, United States

**Abstract** The post-transcriptional mechanisms contributing to molecular regulation of developmental lymphangiogenesis and lymphatic network assembly are not well understood. MicroRNAs are important post-transcriptional regulators during development. Here, we use high throughput small RNA sequencing to identify miR-204, a highly conserved microRNA dramatically enriched in lymphatic vs. blood endothelial cells in human and zebrafish. Suppressing miR-204 leads to loss of lymphatic vessels while endothelial overproduction of miR-204 accelerates lymphatic vessel formation, suggesting a critical positive role for this microRNA during developmental lymphangiogenesis. We also identify the NFATC1 transcription factor as a key miR-204 target in human and zebrafish, and show that NFATC1 suppression leads to lymphatic hyperplasia. The loss of lymphatics caused by miR-204 deficiency can be largely rescued by either endothelial autonomous expression of miR-204 or by suppression of NFATC1. Together, our results highlight a miR-204/NFATC1 molecular regulatory axis required for proper lymphatic development.
DOI: https://doi.org/10.7554/eLife.46007.001

**\*For correspondence:**
flyingfish2@nih.gov

**Competing interests:** The authors declare that no competing interests exist.

## Introduction

The lymphatic system is important for fluid and protein homeostasis, lipid transport, and immunity. Lymphatic malfunction is linked to many pathologies including lymphedema, cancer metastasis, and cardiovascular disease (*Petrova and Koh, 2018*; *Venero Galanternik et al., 2016*; *Alitalo, 2011*; *Stacker et al., 2014*; *Lim et al., 2013*). Primitive lymphatic vessels are derived from lymphatic endothelial cell (LEC) progenitors that transdifferentiate from venous blood endothelial cells (BECs) and subsequently migrate away from the veins to form lymphatic vessels (*Sabin, 1902*; *Hong et al., 2002*; *Koltowska et al., 2013*; *Nicenboim et al., 2015*). A variety of factors have been identified that are required for proper regulation of lymphangiogenesis during development and disease, including PROX1, SOX18, COUPTFII, VEGFC/VEGFR3, NRP2, CCBE1, CLEC2, YAP/TAZ, and NFATC1 (*Wigle and Oliver, 1999*; *Dumont et al., 1998*; *Karkkainen et al., 2004*; *François et al., 2008*; *Srinivasan et al., 2010*; *Yuan et al., 2002*; *Uhrin et al., 2010*; *Hogan et al., 2009*; *Kulkarni et al., 2009*; *Sweet et al., 2015*; *Bui et al., 2016*; *Cho et al., 2019*). Some of our current knowledge regarding development of the lymphatic system has come from research using the zebrafish, a superb model for studying vertebrate organogenesis with optically clear, externally developing embryos ideal for high-resolution live imaging. The development of the zebrafish lymphatic vascular system is highly conserved and stereotyped (*Yaniv et al., 2006*; *Okuda et al., 2012*; *Jung et al., 2017*; *Küchler et al., 2006*), and the availability of transgenic zebrafish expressing fluorescent reporters in lymphatic endothelium makes it straightforward to monitor lymphatic development at the single cell level in vivo and visualize even subtle lymphatic defects using optimized high-resolution microscopy technologies (*Jung et al., 2016*).

MicroRNAs are important posttranscriptional regulators that play crucial roles in developmental, physiological, and disease-related processes in animals (*Gebert and MacRae, 2019*). They are non-coding RNAs approximately 22 nucleotide long that guide Argonaute proteins for gene silencing by mRNA degradation and/or translational repression (*Jonas and Izaurralde, 2015*). Targeting of microRNAs is accomplished by binding of the key nucleotides 2–8 (the 'seed' sequence) and additional microRNA sequences to complementary sequences in target mRNAs (*Bartel, 2009*). The majority of microRNA target sites are located in the 3' untranslated regions (UTRs) of mRNAs, although some microRNA targeting also occurs in 5'UTR and coding sequences (*Tay et al., 2008*; *Lytle et al., 2007*; *Jung et al., 2013*). MicroRNA regulatory networks are thought to confer robustness to biological processes by reinforcing transcription programs and attenuating aberrant transcripts by either switching off or fine-tuning gene expression, helping to buffer against random fluctuations in transcript copy number (*Ebert and Sharp, 2012*; *Staton et al., 2011*; *Choi et al., 2007*). Although we know a great deal about the roles of protein-coding genes during lymphangiogenesis, we still have limited insight into how post-transcriptional mechanisms regulate lymphatic development. Profiling of microRNAs in human lymphatic endothelial cells (LECs) and blood endothelial cells (BECs) has identified BEC-specific microRNAs such as miR-31 and miR-181a that target Prox1 and prevent lymphatic specification in BECs (*Leslie Pedrioli et al., 2010*; *Kazenwadel et al., 2010*; *Dunworth et al., 2014*). These studies suggest some microRNAs can help BECs retain their identity by negatively regulating lymphatic development. Manipulation of endothelial microRNAs has also been shown to result in defective lymphatic development (*Kontarakis et al., 2018*; *Nicoli et al., 2012*; *Chen et al., 2016*). Although these studies have begun to shed light on the role of microRNAs during lymphangiogenesis, our understanding of the role lymphatic microRNAs play during lymphatic vessel development is still limited.

Here, we characterize miR-204, a highly conserved lymphatic-enriched microRNA isolated via small RNA sequencing of human endothelial cells, and demonstrate its critical function during lymphangiogenesis in vivo using the zebrafish. MicroRNA-204-deficient zebrafish display severe defects in lymphatic vessel formation, while excess mir-204 expression in endothelium drives precocious lymphangiogenesis. We identify NFATC1 as a conserved miR-204 target in both human lymphatic endothelial cells and in the zebrafish, with loss of miR-204-mediated silencing resulting in increased NFATC1 transcript levels. As in mammalian lymphatics, attenuating nfatc1 in the zebrafish promotes abnormal lymphatic expansion, and suppressing nfatc1 rescues lymphatic development in mir-204-deficient zebrafish. Our results thus identify a miR-204/NFATC1 molecular pathway critical for lymphatic development.

## Results

### miR-204 is enriched in human and zebrafish lymphatic endothelial cells

We performed small RNA sequencing on human lymphatic endothelial cells (LEC) and human blood endothelial cells (BEC) to identify microRNAs enriched in LECs compared to BECs. We used triplicate samples of total RNA isolated from human dermal lymphatic microvascular endothelial cells (HMVEC-dLy, representing LEC) and human umbilical vein endothelial cells (HUVEC, representing BEC) for sequencing (*Figure 1a*). Prior to sequencing, we used TaqMan quantitative RT-PCR to verify that HMVEC-dLy and HUVEC are enriched for markers representing lymphatic or blood vessel identity, respectively. We tested the expression of lymphatic vascular markers PROX1, FLT4 (also known as VEGFR3), and PDPN, and blood vascular markers NR2F2 (also known as COUP-TFII), KDR (also known as VEGFR2), CDH5 (also known as VE-Cadherin), and EGFL7. HMVEC-dLy express higher levels of PROX1, FLT4, and PDPN, while HUVEC showed enrichment for NR2F2, KDR, CDH5, and EGFL7, showing that these cell types appropriately express genes representative of lymphatic or blood vessel identity (*Figure 1—figure supplement 1*). A total of ~10 million reads were collected for each RNAseq sample and the sequences were aligned using the miRbase v22 (*Kozomara and Griffiths-Jones, 2014*). We excluded from further analysis any microRNAs that were represented by less than 10 reads in three or more of the six (two triplicate) sequenced samples, resulting in 445 annotated microRNAs. 98 of these microRNAs showed a significant difference between the LEC and BEC samples ($p<0.01$, false discovery rate (FDR) $< 0.01$). The normalized sequencing data have been submitted to the Gene Expression Omnibus repository with accession number GSE126679. From

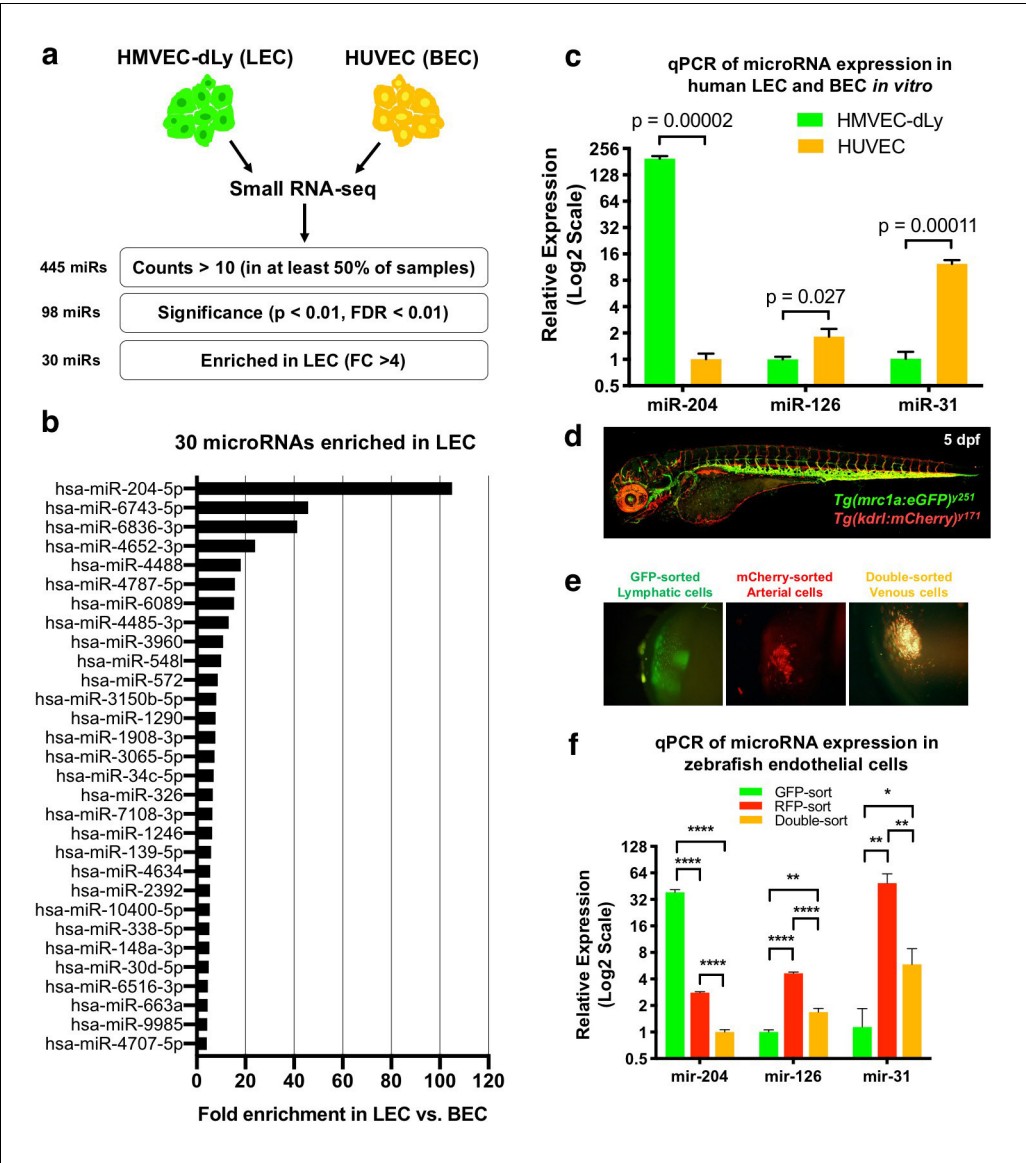

**Figure 1.** Identification of lymphatic microRNAs enriched in human and zebrafish lymphatic endothelial cells. (**a**) Schematic diagram of the workflow for small RNA sequencing from lymphatic (HMVEC-dLy) and blood (HUVEC) endothelial cells and selection of microRNAs enriched in lymphatic endothelial cells. (**b**) Relative fold enrichment of the 22 most highly enriched microRNAs in LEC versus BEC small RNA sequence data (average of triplicate samples from each group). (**c**) Quantitative TaqMan RT-PCR measurement of the relative expression of three different microRNAs in HMVEC-dLy (LEC) and HUVEC (BEC). Levels of mir-204 are normalized to HUVEC (BEC) levels, while levels of mir-126 and mir-31 are normalized to HMVEC-dLy (LEC) levels. Three biological replicates were analyzed. (**d**) Confocal image of a five dpf *Tg(mrc1a:eGFP)[y251]*, *Tg(kdrl:mCherry)[y171]* double-transgenic larva (lateral view, rostral to the left). (**e**) Confocal images of lymphatic (GFP-positive), arterial (mCherry-positive), and venous (GFP and mCherry double-positive) endothelial cell pellets isolated from dissociated five dpf transgenic animals such as that in panel d by Fluorescence Activated Cell Sorting (FACS). (**f**) Quantitative TaqMan RT-PCR measurement of the relative expression of mature mir-204, mir-126, and mir-31 in FACS-sorted zebrafish endothelial cells. FACS-sorted cells from ~1000 5 dpf larvae were used and three technical replicates were analyzed. Levels of mir-204 are normalized to venous (GFP and mCherry double-positive) levels, while levels of mir-126 and mir-31 are normalized to lymphatic (GFP-positive) levels. All graphs are analyzed by t-test and the mean ± standard deviation (SD) is shown. *, p<0.05; **, p<0.01; ****, p<0.0001.
DOI: https://doi.org/10.7554/eLife.46007.002

The following source data and figure supplements are available for figure 1:

**Source data 1.** List of microRNAs from the small RNA sequencing data with numeric values comparing LEC vs. BEC.

*Figure 1 continued on next page*

*Figure 1 continued*

DOI: https://doi.org/10.7554/eLife.46007.005

**Source data 2.** Numerical data for *Figure 1* and *Figure 1—figure supplement 1*.

DOI: https://doi.org/10.7554/eLife.46007.006

**Figure supplement 1.** Differential expression of vascular genes in HMVEC-dLy and HUVEC.

DOI: https://doi.org/10.7554/eLife.46007.003

**Figure supplement 2.** Trunk vascular patterning and vascular gene expression in FACS-sorted endothelial cells from transgenic zebrafish.

DOI: https://doi.org/10.7554/eLife.46007.004

the 98 microRNAs we identified 30 that were highly enriched in LECs (fold change (FC) >4) and 20 that were highly enriched in BECs (FC >4) (*Figure 1a* and *Figure 1—source data 1*). Among the 30 LEC-enriched microRNAs, miR-204–5 p was by far the most highly enriched in LEC, with 105-fold higher representation in the LEC sequences compared to BEC (*Figure 1b*). Using the TaqMan micro-RNA qPCR assay, we confirmed that miR-204–5 p is very highly enriched in LEC compared to BEC, in contrast to previously reported vascular microRNAs miR-126 and miR-31 that are more highly enriched in BEC (*Figure 1c*).

We used the zebrafish as an in vivo system to examine the role of mir-204 during lymphatic development, beginning by testing whether zebrafish mir-204 is also enriched in LEC. Zebrafish have a conserved, highly stereotyped developing lymphatic vascular network that is readily visualized using transgenic reporter lines (*Okuda et al., 2012*; *Jung et al., 2017*) (*Figure 1—figure supplement 2a, b*). We used 5 day post-fertilization (dpf) *Tg(mrc1a:eGFP)^{y251}*, *Tg(kdrl:mCherry)^{y171}* double-transgenic larvae with EGFP-positive lymphatic EC, mCherry-positive arterial EC, and EGFP and mCherry double-positive venous EC (*Figure 1d*) to isolate each of these endothelial cell populations by fluorescence activated cell sorting (FACS). Transgenic larvae were dissociated into single cells and subjected to FACS sorting to obtain EGFP-sorted lymphatic cells, mCherry-sorted arterial cells, and double-sorted venous cells (*Figure 1e*). The EGFP-sorted lymphatic cells showed strong expression of lymphatic markers lyve1b and prox1a, while arterial and venous endothelial cells expressed relatively low levels of these transcripts (*Figure 1—figure supplement 2c*). In contrast, blood vascular markers kdrl and cdh5 were more abundant in mCherry-sorted arterial and double-sorted venous cells compared to EGFP-sorted lymphatic cells (*Figure 1—figure supplement 2d*). Using these sorted endothelial cell populations we were able to show that zebrafish mir-204 is highly enriched in EGFP-sorted lymphatic cells, while the blood endothelial microRNAs mir-126 and mir-31 are more enriched in mCherry-positive or double-positive BECs (*Figure 1f*). These data show that miR-204 is a highly conserved microRNA enriched in the lymphatic endothelium in both humans and zebrafish.

## Developmental lymphangiogenesis is suppressed by miR-204 deficiency

Human miR-204 is located in the sixth intron of TRPM3 (Transient Receptor Potential Cation Channel Subfamily M Member 3), and the mature microRNA is 100% conserved amongst a variety of vertebrate species (*Figure 2—figure supplement 1a*). The zebrafish genome contains three paralogues of mir-204 (miRBase.org). As in the human genome, one mir-204 is found in intron 5 of the zebrafish *trpm3* gene (*mir-204–1*) (*Figure 2—figure supplement 1a*). However, additional zebrafish mir-204 sequences are also found in intron 5 of *trpm1a* and in intron 4 of *trpm1b* (*mir-204–2* and *mir-204–3*, respectively; *Figure 2—figure supplement 1b*). The single human *TRPM1* gene contains a closely related paralogue microRNA, miR-211, in intron 6. Although each of the three copies of mir-204 in the zebrafish has unique precursor sequences, their mature mir-204 sequences are 100% identical (*Figure 2a*). To determine the function of mir-204 in lymphatic vessel formation during early development, we began by using morpholino (MO) antisense oligomers to target and block the function of endogenous mir-204 in zebrafish. Knocking down microRNA function using MOs is an excellent targeting strategy for microRNAs because unlike protein-coding genes the functional products of microRNA genes are RNAs (*Flynt et al., 2017*). Four different MOs were designed to use different strategies to block mir-204 function (*Figure 2a*). The pan-204 MO targets the mature mir-204 sequence and knocks down all mir-204s. Precursor hairpin structures must be properly cleaved by Dicer to permit maturation of functional microRNAs (*Park et al., 2011*; *Flynt et al., 2017*; *Kloosterman et al., 2007*). We designed specific MOs targeting the dicer-cleavage sites of *mir-204–*

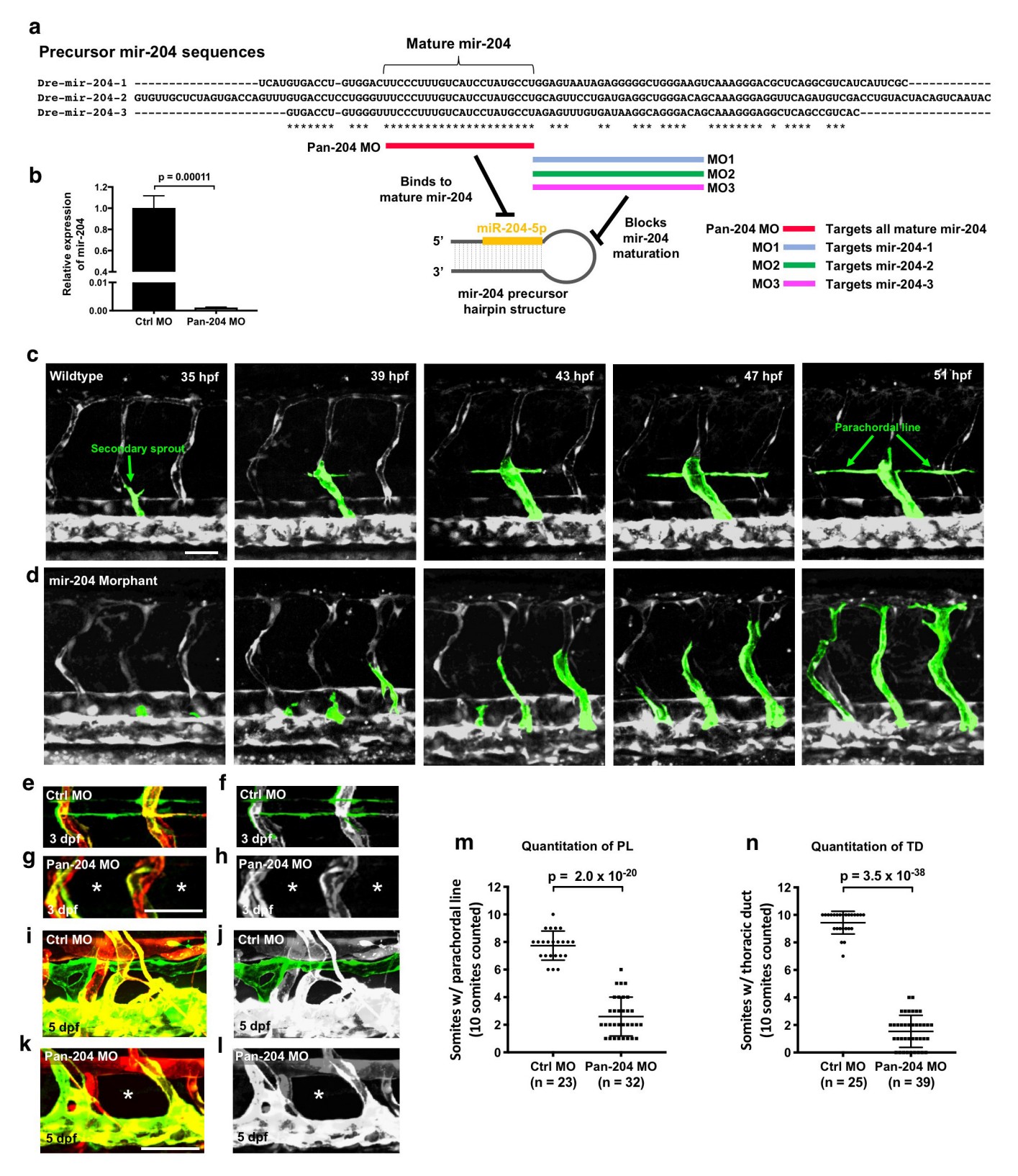

**Figure 2.** Defective lymphangiogenesis in mir-204 deficient zebrafish. (a) Sequence alignment of the three zebrafish precursor mir-204 sequences (*mir-204–1*, *mir-204–2*, and *mir-204–3*) and a schematic diagram showing four morpholinos (pan-204 MO, MO1, MO2, and MO3) targeting them. The data

*Figure 2 continued on next page*

*Figure 2 continued*

shown in the rest of this figure (panels b-m) uses the pan-204 MO targeting the mature mir-204 sequence generated by all three zebrafish mir-204 loci. (b) Quantitative TaqMan RT-PCR measurement of the relative levels of mature miR-204 in one dpf control MO- or pan-204 MO-injected embryos, normalized to controls. Three biological replicates were analyzed. (c,d) Time series of confocal images of trunk vessels in 35–51 hpf $Tg(mrc1a:eGFP)^{y251}$; $Tg(kdrl:mCherry)^{y171}$ control (c) or pan-204 morphant (d) animals, with secondary sprouts highlighted in green. (e–h) Confocal images of the parachordal line in three dpf $Tg(mrc1a:eGFP)^{y251}$;$Tg(kdrl:mCherry)^{y171}$ animals injected with either control MO (e, f) or pan-204 MO (g, h). In panel f the parachordal line is highlighted in green and other vessels are in gray. The absence of the parachordal line is noted with asterisks in panels g and h. (i–l) Confocal images of the thoracic duct in five dpf $Tg(mrc1a:eGFP)^{y251}$;$Tg(kdrl:mCherry)^{y171}$ animals injected with either control MO (i, j) or pan-204 MO (k, l). In panel j the thoracic duct is highlighted in green and other vessels are in gray. The absence of the thoracic duct is noted with asterisks in panels k and l. (m) Quantification of parachordal line formation in three dpf animals injected with either control MO (n = 23) or pan-204 MO (n = 32). The same 10 somitic segments were scored in each animal for the presence or absence of an intact parachordal line. (n) Quantification of thoracic duct formation in five dpf animals injected with either control MO (n = 25) or pan-204 MO (n = 39). The same 10 somitic segments were scored in each animal for the presence or absence of an intact thoracic duct. All images are lateral views. Scale bar: 50 µm (c, g, k). All graphs are analyzed by t-test and the mean ± SD is shown.

DOI: https://doi.org/10.7554/eLife.46007.007

The following video, source data, and figure supplements are available for figure 2:

**Source data 1.** Numerical data for *Figure 2* and *Figure 2—figure supplement 2*.
DOI: https://doi.org/10.7554/eLife.46007.010

**Figure supplement 1.** Evolutionarily conservation of miR-204.
DOI: https://doi.org/10.7554/eLife.46007.008

**Figure supplement 2.** The effects of single or combined injections of mir-204 targeting MOs.
DOI: https://doi.org/10.7554/eLife.46007.009

**Figure 2—video 1.** Time lapse video of wildtype (top) and pan204 morphant (bottom) vessel development.
DOI: https://doi.org/10.7554/eLife.46007.011

1, *mir-204–2*, and *mir-204–3*, respectively, to individually block the maturation of each of the three precursor mir-204 sequences (*Figure 2a*). Injection of 0.5 ng of pan-204 MO led to highly efficient suppression of mature mir-204 levels (*Figure 2b*). At the 0.5 ng dose there were no noticeable morphological anomalies, although slightly higher doses (0.75 ng) resulted in some pericardial edema, jaw defects, mild microcephaly and reduced eye development (*Figure 2—figure supplement 1c*). To avoid potential secondary effects on lymphatics caused by these other abnormalities, we used the 0.5 ng dose for all experiments with the pan-204 MO. At around 36 hpf, vascular 'secondary sprouts' emerge from the cardinal vein, migrate dorsally, pause half-way up the trunk, and then contribute to formation of the parachordal lines (an early transient lymphatic progenitor structure) along the horizontal myoseptum at 2–3 dpf. From 3–5 dpf dorsal and ventral sprouts emerge from the parachordals to give rise to trunk lymphatic network (*Figure 1—figure supplement 2a*), including the intersegmental lymphatic vessels, dorsal longitudinal lymphatic vessels (DLLV), and thoracic duct (*Cha et al., 2012*; *Yaniv et al., 2006*; *Hogan et al., 2009*). Consistent with previous observations, the secondary sprouts in wild type control animals migrated dorsally to the horizontal myoseptum, turned laterally, and formed the parachordal lines (*Figure 2c* and *Figure 2—video 1*). Although initial secondary sprout formation and dorsal growth was normal in mir-204-deficient animals, secondary sprouts in these animals failed to stop at the horizontal myoseptum and form the parachordal line, but instead continued to grow dorsally and contributed to veins (*Figure 2d* and *Figure 2—video 1*). Examination of the parachordal line at three dpf (*Figure 2e–h,m*) showed that compared to control MO-injected animals that had parachordal lines in most somitic segments at this stage (*Figure 2e,f,m*), pan-204 MO-injected animals formed parachordals in only about 2 segments per 10 somites (*Figure 2g,h,m*). Similarly, examination of the thoracic duct at five dpf (*Figure 2i–l,n*) showed that control MO-injected animals had a thoracic duct in virtually all segments (*Figure 2i,j,n*) while mir-204-deficient animals displayed nearly complete loss of thoracic duct formation (*Figure 2k,l,n*).

## mir-204 function is required for lymphatic development

To independently confirm our pan-204 MO findings and further examine the relative importance of individual zebrafish mir-204s, we carried out experiments using the MOs independently targeting the dicer-cleavage sites of each of the three zebrafish mir-204s (*Figure 2a* and *Figure 2—figure supplement 2*). The sequences targeted by these MOs (MO1, MO2, MO3) differ from one another

by 27–46% (*Figure 2—figure supplement 2a*). Injection of 0.5 ng of each individual MO alone did not affect lymphatic vessel development (*Figure 2—figure supplement 1b–d,i*). Pairwise co-injection of 0.5 ng MO1 and 0.5 ng MO2 resulted in defective thoracic duct formation (*Figure 2—figure supplement 2e,j*), but pairwise co-injection of either MO1 + MO3 or MO2 + MO3 did not affect lymphatic vessel formation (*Figure 2—figure supplement 2f,g,j*). Animals injected with 0.5 ng of each of the three MOs (MO1, MO2, and MO3, for a total of 1.5 ng injected) had lymphatic defects similar to but no more severe than animals injected with either the MO1 + MO2 combination or 0.5 ng of pan-204 MO (*Figure 2—figure supplement 2h,j* and *Figure 2n*). These results suggest that *mir-204–1* and *mir-204–2* play relatively more important roles than *mir-204–3* during lymphatic development, and confirm that suppressing mir-204 function leads to defective lymphatic vessel formation.

To further examine the role of mir-204 in lymphatic development, we generated genetic mutants using the Clustered Regularly Interspaced Short Palindromic Repeats (CRISPR)/Cas9 and CRISPR/Cpf1 systems. Using an active single guide RNA (sgRNA) targeting *mir-204–1* (*Figure 3a*), we isolated a 22 bp CRISPR/Cas9 *mir-204–1* deletion mutant that deletes the majority of the mature mir-204 sequence (17 out of 22 nt) and the entire seed sequence at this locus (*Figure 3b*). Only an approximately 20% decrease in mir-204 levels in *mir-204–1* homozygous mutants (*miR-204–1$^{-/-}$*) compared to wildtype embryos suggests that reduction from *mir-204–1* loss is compensated for by *mir-204–2* and/or *mir-204–3* (*Figure 3—figure supplement 1a*). The *mir-204–1$^{-/-}$* animals were viable and fertile and did not have obvious morphological defects up to adulthood (*Figure 3c,d* and data not shown). Entirely consistent with the data generated exclusively using morpholinos (*Figure 2—figure supplement 2*), *mir-204–1$^{-/-}$* mutant animals injected with either MO2 (*Figure 3e,h*) or with MO2+MO3 (*Figure 3g,h*) fail to properly form the thoracic duct and other lymphatic vessels, but *mir-204–1$^{-/-}$* mutants injected with MO3 developed normal lymphatics (*Figure 3f,h*). MO2-injected *mir-204–1$^{-/-}$* mutant animals also showed defects in earlier parachordal line formation (*Figure 3—figure supplement 1b–d*), like pan-204 MO-injected animals (*Figure 2g,h,m*). These data indicate that suppressing biogenesis of mir-204 from the *mir-204–1* and *mir-204–2* loci is sufficient to disrupt mir-204 function in lymphatic vessel formation, regardless of whether a genetic mutant or morpholino is used to suppress *miR-204–1*.

We also used CRISPR/Cpf1 targeting to generate −11 bp and −17 bp deletion mutations for *mir-204–2* and a −12 bp deletion for *mir-204–3* (*Figure 3—figure supplement 2a*). Since *mir-204–1$^{-/-}$* animals are viable and fertile, we co-injected two separate sgRNAs targeting *mir-204–2* and *mir-204–3*, respectively, into *mir-204–1$^{-/-}$* embryos to facilitate generation of *mir-204* triple mutants. We obtained triple mutants deleting the seed sequences of all three mir-204s (*Figure 3—figure supplement 2a*). Although expression of mir-204 was strongly reduced in *mir-204* triple homozygous mutants (*Figure 3—figure supplement 2b*), they did not display statistically significant lymphatic defects (*Figure 3—figure supplement 2c–e*), presumably due to an unknown compensatory mechanism taking place in animals where all genomic mature *mir-204* sequences are defective.

## Endothelial expression of mir-204 drives precocious lymphangiogenesis

To further confirm and examine the role of mir-204 in lymphatics, we used the mrc1a promoter (*Jung et al., 2017*) to drive mir-204 expression in venous and lymphatic endothelial cells using a previously described transgene vector for microRNA expression (*Nicoli et al., 2010*) containing splice donor (SD) and splice acceptor (SA) sequences from the EF1alpha gene, with the *dre-mir-204–1*-containing trpm3 intron five sequence cloned in between them (*Figure 4a*). As a control, we introduced a 4 bp mismatch into the seed sequence position 2–5 (TCCC to AGGG) using site-directed mutagenesis (*Figure 4b*). We injected *mrc1a:mir204(4 bp_seed_mut)-eGFP* (as control) or *mrc1a:mir204-eGFP* vector DNA into single-cell wild type embryos and analyzed the mosaic contribution to thoracic duct formation at three dpf, when the cells from parachordal line start to migrate ventrally to form thoracic duct. In comparison to control embryos that were just beginning thoracic duct formation (*Figure 4b*), embryos injected with *mrc1a:mir204-eGFP* formed a significantly increased number of thoracic duct segments (*Figure 4c*) suggesting that endothelial expression of mir-204 promotes early thoracic duct development (*Figure 4d*). We also isolated germline transgenic animals carrying this *Tg(mrc1a:mir204-egfp)* transgene. Similar to the results from the mosaic analysis, germline transgenic progeny generated from these fish display precocious development of the thoracic duct (*Figure 4—figure supplement 1a–c*). These animals are viable and fertile with no obvious

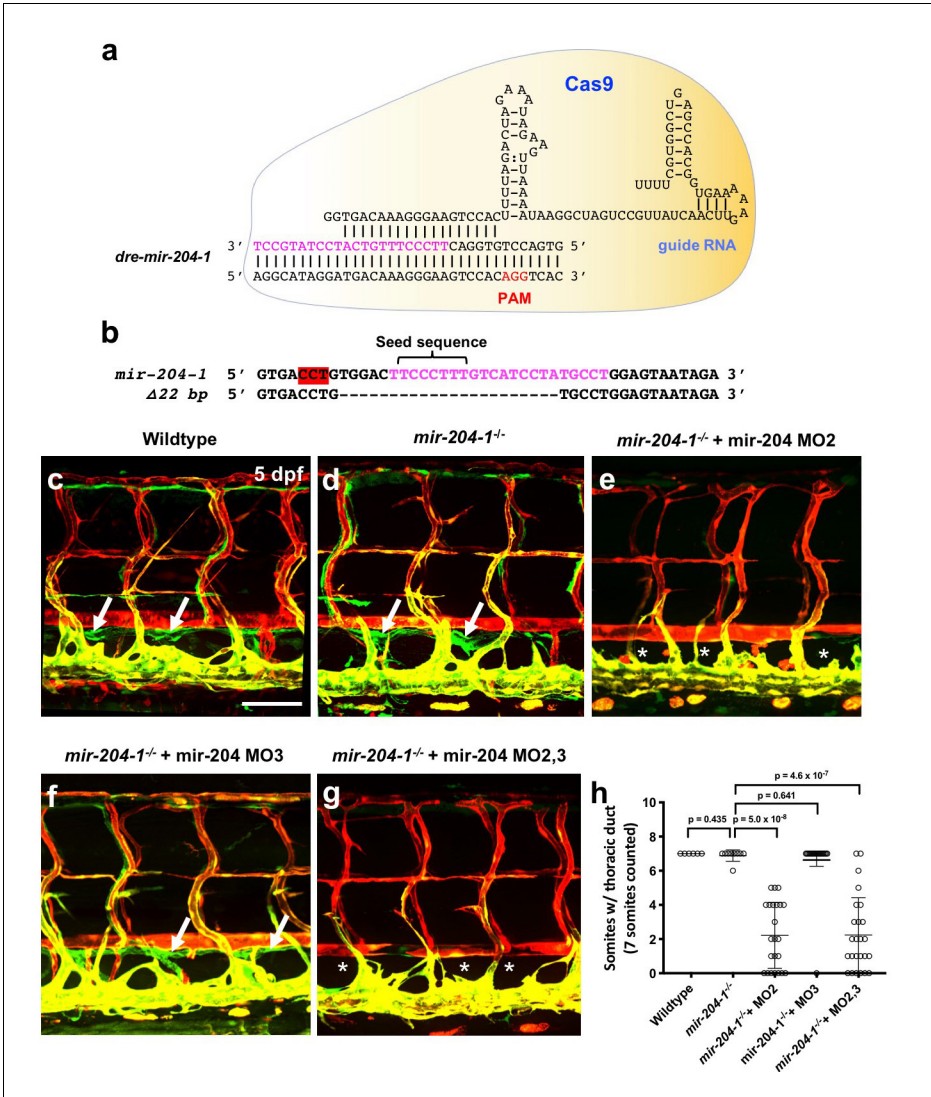

**Figure 3.** Mir-204 function is required for lymphatic development. (a) Schematic of CRISPR/Cas9 and guide RNA targeting of *mir-204–1*. (b) Sequence alignment of wildtype and *mir-204–1* mutant genomic DNA. The mature mir-204 sequence is noted in magenta, and the PAM sequence is highlighted in red (on the reverse strand). The mutant carries 22 bp deletion that removes 17 nucleotides of the mature mir-204 sequence. (c–g) Representative confocal images of the mid-trunk of 5 dpf wild type sibling (c), *mir-204–1*[-/-] mutant (d), MO2-injected *mir-204–1*[-/-] mutant (e), MO3-injected *mir-204–1* mutant (f), and MO2 + MO3 co-injected *mir-204–1*[-/-] mutant (g) animals. Images are lateral views of *Tg(mrc1a:eGFP)*[y251], *Tg(kdrl:mCherry)*[y171] double-transgenic animals, rostral to the left. The thoracic duct is labeled with white arrows, and absence of the thoracic duct is noted with asterisks. (h) Quantification of thoracic duct formation in five dpf wild type (n = 6), *mir-204–1*[-/-] mutant (n = 9), MO2-injected *mir-204–1*[-/-] mutant (n = 23), MO2-injected *mir-204–1*[-/-] mutant (n = 19), and MO2 + MO3 co-injected *mir-204–1*[-/-] mutant animals (n = 25). The number of somitic segments with an intact thoracic duct was counted, with the same seven mid-trunk somites measured in each animal. Scale bar: 100 μm (c). All graphs are analyzed by t-test and the mean ± SD is shown.

DOI: https://doi.org/10.7554/eLife.46007.012

The following source data and figure supplements are available for figure 3:

**Source data 1.** Numerical data for *Figure 3* and *Figure 3—figure supplement 1* and *Figure 3—figure supplement 2*.

DOI: https://doi.org/10.7554/eLife.46007.015

**Figure supplement 1.** Lymphatic differentiation defects in mir-204-deficient animals.

DOI: https://doi.org/10.7554/eLife.46007.013

**Figure supplement 2.** MicroRNA 204 mutants.

*Figure 3 continued on next page*

*Figure 3 continued*

DOI: https://doi.org/10.7554/eLife.46007.014

morphological abnormalities or other defects, including normal stage-specific spacing between the dorsal aorta and posterior cardinal vein (*Figure 4—figure supplement 1d*), despite an approximately 2-fold increase in mature miR-204 levels measured by qPCR at three dpf (*Figure 4—figure supplement 1e*). These results suggest that increased endothelial mir-204 expression specifically promotes early lymphatic development.

To further confirm the endothelial cell autonomous role of miR-204 in lymphatic development, we examined whether endothelial specific miR-204 expression could also rescue the loss-of-lymphatic phenotype in mir-204-deficient animals. Since the transgene construct was generated using *mir-204–1* genomic sequence, MO2 (which specifically targets mir-204–2) does not affect its function (*Figure 2* and *Figure 2—figure supplement 2a*). Combined loss of mir-204–1 and mir-204–2 is sufficient for the full miR-204 lymphatic phenotype (*Figure 2—figure supplement 2*, *Figure 3*, *Figure 4e*). MO2-injected *mir-204–1*$^{-/-}$ embryos showed strong suppression of thoracic duct formation (*Figure 4e,h,k*), but this was rescued by co-injection of the mrc1a:miR204-eGFP transgene (*Figure 4f,g,i,j,k*), with stronger 'rescue' noted in animals with a higher mosaic contribution from the transgene (*Figure 4g,j,k*). Together, these data indicate that endothelial-autonomous miR-204 function is required for developmental lymphangiogenesis.

## NFATC1 is a conserved target of miR-204

Based on our observation that this microRNA plays an important role during lymphatic development, we began a search for potential miR-204 target genes required for developmental lymphangiogenesis. We (i) began with a list of human genes previously implicated in lymphatic development, then (ii) bioinformatically identified which genes in this set had potential miR-204 target sites using TargetScan (*Agarwal et al., 2015*), and then (iii) bioinformatically identified which of these genes also had corresponding zebrafish orthologs with potential miR-204 target sites (in order to ensure that we were looking at key, conserved targets that we could functionally study in both human cell culture and in zebrafish (*Supplementary file 1*). The Nuclear Factor of Activated T Cells 1 (NFATC1) gene immediately came to our attention as a strong candidate meeting these criteria.

NFATC1 is expressed in developing lymphatic endothelial cells and genetic ablation of Nfatc1 in mice causes abnormal lymphatic vessel patterning and lymphatic hyperplasia (*Norrmén et al., 2009*). Human NFATC1 contains a putative miR-204 binding site in its 3'UTR (*Figure 5a*). The NFATC1 3'UTR was cloned downstream of a luciferase reporter and this plasmid DNA construct was co-transfected into HEK293 cells together with either miR-204 or (as a negative control) miR-126. Luciferase reporter activity was significantly suppressed in the presence of miR-204 but not in the presence of the miR-126 control (*Figure 5b*). To confirm the specificity of NFATC1 miR-204 target site recognition, we mutated four nucleotides in the seed binding sequence of the NFATC1 3'UTR (*Figure 5a*) and demonstrated that this rendered the construct insensitive to suppression by miR-204 (*Figure 5b*), confirming that this binding site is critical for direct targeting. Overexpression of miR-204 in human LECs suppressed endogenous NFATC1 expression while miR-204 inhibitor (antagomir) increased endogenous NFATC1 expression, showing that miR-204 also regulates endogenous NFATC1 transcript levels in human LEC (*Figure 5c*). We observed similar regulation of nfatc1 by mir-204 in the zebrafish. Zebrafish *nfatc1* also contains a mir-204 binding site in its 3'UTR (*Figure 5d*). The activity of a luciferase reporter containing the zebrafish *nfatc1* 3'UTR is also strongly suppressed by miR-204, and this suppression is fully rescued by mutating the seed binding sequence in the nfatc1 3'UTR (*Figure 5e*). Importantly, mir-204 knockdown using the pan-204 MO (targeting all three zebrafish mir-204s) results in increased levels of endogenous nfatc1 transcript in five dpf zebrafish embryos (*Figure 5f*). Together, these data demonstrate conserved miR-204-mediated regulation of NFATC1.

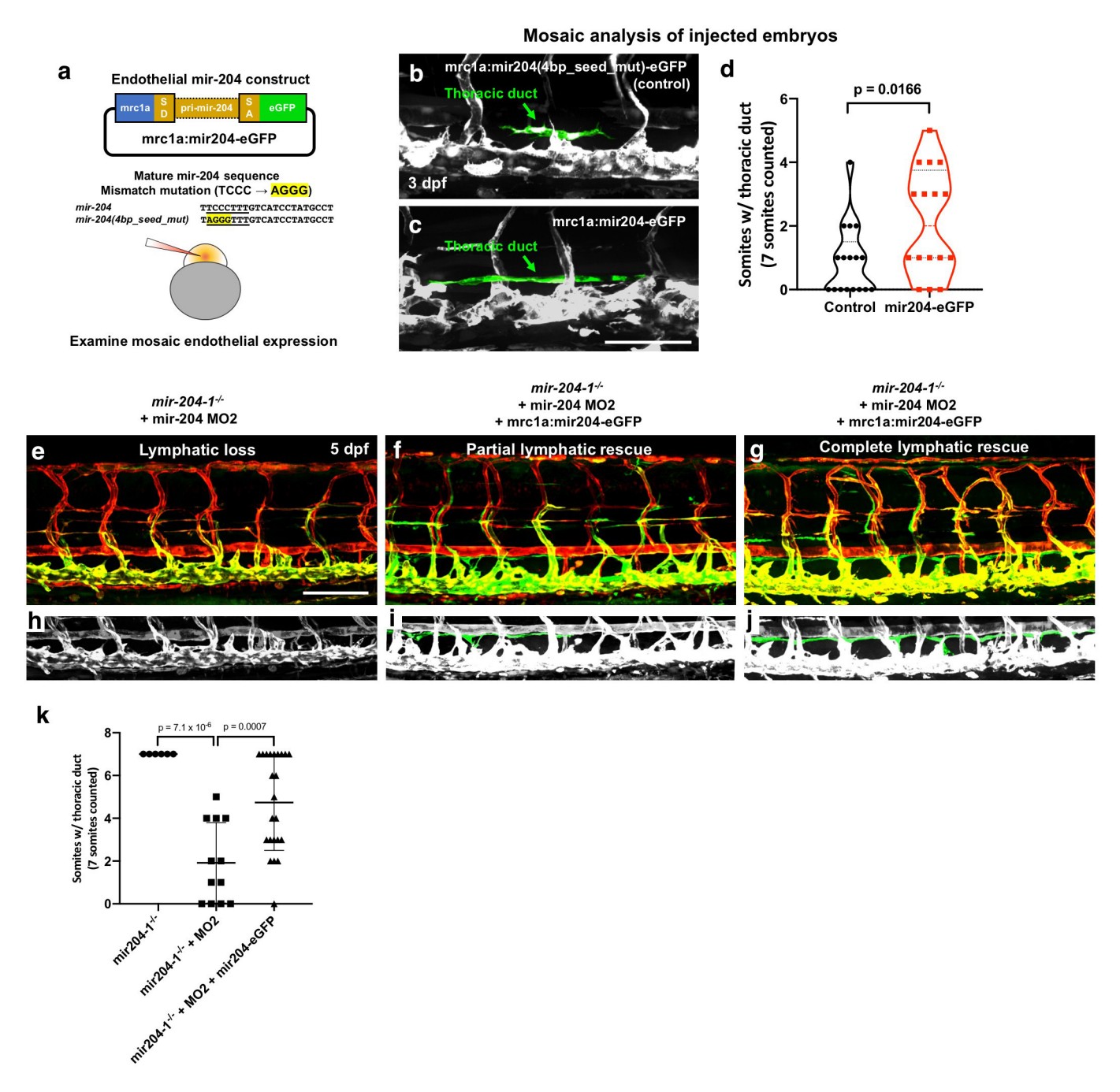

**Figure 4.** Endothelial expression of mir-204 drives precocious lymphatic development and rescues the loss-of-lymphatic phenotype in mir-204-deficient animals. (a) Schematic illustration of the mir-204 expression construct. The EGFP expression cassette is driven by the *mrc1a* promoter (*Jung et al., 2017*), with ~1 kb of *dre-miR-204–1* genomic sequence from the fifth intron of the *trpm3* gene cloned between splice donor (SD) and splice acceptor (SA) sequences and flanking exonic sequences from the *ef1a* gene. The original vector backbone was previously described (*Nicoli et al., 2010*). As a control, a 4 bp mismatch mutation was introduced in the seed (underline) sequence of mature mir-204 sequence. The construct was injected into 1 cell stage embryos to examine the mosaic endothelial expression. (b,c) Representative confocal images of mid-trunk of 3 dpf embryos injected with (b) control *mrc1a:mir204(4 bp_seed_mut)-eGFP* DNA or (c) *mrc1a:mir204-eGFP* DNA. The thoracic duct is pseudocolored in green, with other vessels in gray. (d) Quantification of thoracic duct formation in animals injected with either *mrc1a:eGFP* control or *mrc1a:mir204-eGFP* DNA. (e–g) Confocal images of *Tg(mrc1a:eGFP)^y251, Tg(kdrl:mCherry)^y171* double-transgenic MO2-injected *mir-204–1^-/-* mutant animals without (e) or with (f,g) co-injected *mrc1a:mir204-eGFP* DNA. (h–j) Cropped portions of the corresponding images in e-g, with the thoracic duct pseudocolored in green and other nearby vessels in gray. (k) Quantification of thoracic duct formation in animals as in panels e-g. The number of somitic segments with an intact thoracic duct

*Figure 4 continued on next page*

*Figure 4 continued*

was counted, with the same seven mid-trunk somites measured in each animal. Scale bar: 100 μm (**c,e**). All graphs are analyzed by t-test and the mean ± SD is shown.

DOI: https://doi.org/10.7554/eLife.46007.016

The following source data and figure supplement are available for figure 4:

**Source data 1.** Numerical data for *Figure 4* and *Figure 4—figure supplement 1*.

DOI: https://doi.org/10.7554/eLife.46007.018

**Figure supplement 1.** Germline endothelial expression of mir-204 drives precocious lymphatic development.

DOI: https://doi.org/10.7554/eLife.46007.017

## Suppression of NFATC1 causes thoracic duct hyperplasia in the zebrafish

As noted above, loss of Nfatc1 in mice is associated with lymphatic hyperplasia (*Norrmén et al., 2009*). Using two different morpholinos targeting *nfatc1* (splice MO and ATG MO), we show that knockdown of this gene in the zebrafish also causes thoracic duct enlargement, similar to that shown in mice, with no noticeable abnormal blood vessel formation (*Figure 6a–e* and *Figure 6—figure supplement 1a–e*). The calcineurin inhibitor cyclosporin A (CsA) blocks signaling downstream from

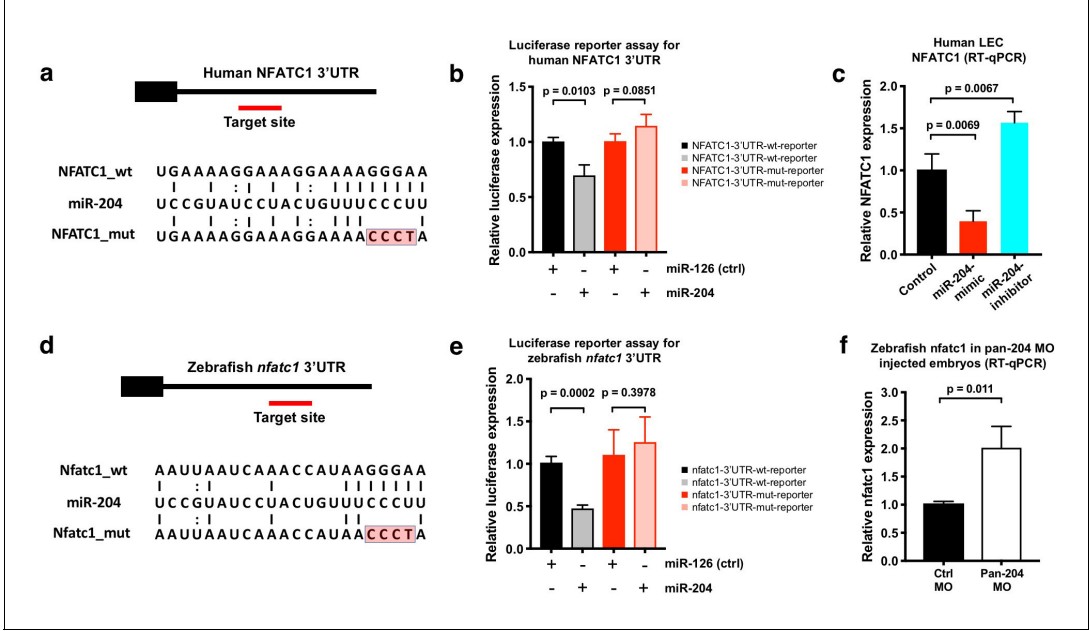

**Figure 5.** NFATC1 is a conserved target of miR-204. (**a**) Sequence alignment of mature miR-204 (middle line) and its target region in the human NFATC1 3'UTR (top line). A mutant form of the human NFATC1 3'UTR used for the luciferase assay in panel b is also shown (bottom line; four mismatches in the seed binding region are highlighted in red). (**b**) Quantitative luciferase reporter assay using wild type or mutant forms of the human NFATC1 3'UTR transfected into HEK293 cells together with either miR-204 or miR-126 (control). Four biological replicates were analyzed. (**c**) Quantitative TaqMan RT-PCR measurement of relative endogenous NFATC1 transcript levels in human LEC (HMVEC-dLy) transfected with miR-204-mimic or miR-204-antagomir, normalized to control mock transfected levels. (**d**) Sequence alignment of mature miR-204 (middle line) and its target region in the zebrafish nfatc1 3'UTR (top line). A mutant form of the zebrafish nfatc1 3'UTR used for the luciferase assay in panel e is also shown (bottom line; four mismatches in the seed binding region are highlighted in red). (**e**) Quantitative luciferase reporter assay using wildtype or mutant forms of the zebrafish nfatc1 3'UTR co-transfected into HEK293 cells together with either miR-204 or miR-126 (control). Four biological replicates were analyzed. (**f**) Quantitative TaqMan RT-PCR measurement of relative endogenous zebrafish nfatc1 transcript levels in five dpf animals that were injected with either control MO or pan-204 MO. Three biological replicates were analyzed. All graphs are analyzed by t-test and the mean ± SD is shown.

DOI: https://doi.org/10.7554/eLife.46007.019

The following source data is available for figure 5:

**Source data 1.** Numerical data for *Figure 5*.

DOI: https://doi.org/10.7554/eLife.46007.020

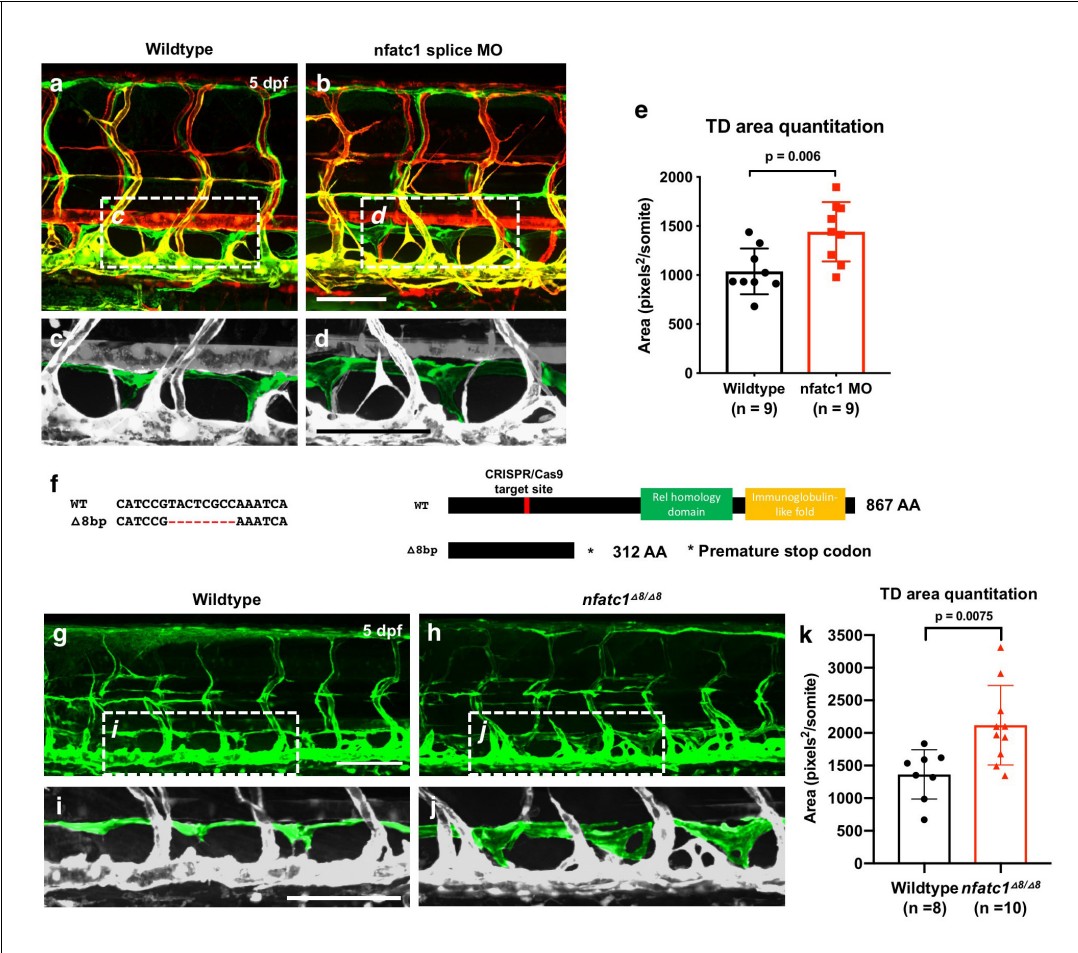

**Figure 6.** Suppression of Nfatc1 promotes enlargement of the thoracic duct. (**a,b**) Confocal images of the mid-trunk of 5 dpf *Tg(mrc1a:eGFP)^y251^, Tg (kdrl:mCherry)^y171^* double-transgenic control MO (**a**) or nfatc1 splice MO (**b**) injected animals. The dashed boxes in panels a and b show the areas magnified in panels c and d, respectively. (**c,d**) Magnified images from panels a and b, with the thoracic duct pseudocolored in green and other vessels in gray. (**e**) Quantitation of thoracic duct size measured as the area encompassed by the thoracic duct in confocal images of the same seven mid-trunk somitic segments in five dpf wildtype (n = 9) and nfatc1 MO-injected (n = 9) animals. (**f**) Sequence alignment of wildtype and *nfatc1^△8/△8^* mutant genomic DNA. Schematic of nfatc1 protein domains, CRISPR target site, and truncated mutant nfatc1 polypeptides. (**g,h**) Confocal images of the mid-trunk of 5 dpf *Tg(mrc1a:eGFP)^y251^* wildtype (**g**) or *nfatc1^△8/△8^* mutant (**h**) animals. (**i,j**) Magnified images from panels g and h, with the thoracic duct pseudocolored in green and other vessels in gray. (**k**) Quantitation of thoracic duct size measured as the area encompassed by the thoracic duct in confocal images of the same seven mid-trunk somitic segments in five dpf wildtype (n = 8) and *nfatc1^△8/△8^* mutant (n = 10) animals. Rostral is to the left in all images. Scale bar = 100 μm (**b,d,g,i**). All graphs are analyzed by t-test and the mean ± SD is shown.
DOI: https://doi.org/10.7554/eLife.46007.021

The following source data and figure supplement are available for figure 6:

**Source data 1.** Numerical data for *Figure 6* and *Figure 6—figure supplement 1*.
DOI: https://doi.org/10.7554/eLife.46007.023
**Figure supplement 1.** Suppressing nfatc1 promotes thoracic duct enlargement.
DOI: https://doi.org/10.7554/eLife.46007.022

NFAT, and treatment of mice with CsA during embryonic stages phenocopies the lymphatic effects caused by genetic ablation of Nfatc1 (*Norrmén et al., 2009*). The zebrafish larvae treated with a low doses of CsA (1 ug/mL) also displayed thoracic duct enlargement phenotype observed in nfatc1 MO-injected animals (*Figure 6—figure supplement 1f–j*). We used CRISPR/Cas9 mutagenesis to generate −8 bp frameshift mutation in the *nfatc1* gene with premature stop codons producing truncated nfatc1 polypeptides lacking key functional domains (*Figure 6f*). Like nfatc1 morphants and CsA-treated animals, *nfatc1^△8/△8^* mutants also had enlarged thoracic ducts at five dpf (*Figure 6g–*

*k*). Therefore, consistent with previous data in mice, our results suggest that nfatc1 is required for proper lymphatic development in the zebrafish.

## Suppression of nfatc1 rescues lymphatic development in mir-204-deficient zebrafish

The results above suggest that increased expression of nfatc1 might be at least partially responsible for the defects in lymphatic vessel development in mir-204-deficient animals. To test this idea, we examined whether lymphatic vessel formation in mir-204-deficient animals could be 'rescued' by nfatc1 knockdown. As already described above, pan-204 MO-injected animals fail to form the thoracic duct (*Figure 7a,b,d,e,j*) and dorsal longitudinal lymphatic vessel (DLLV) in the trunk (*Figure 7a, b,g,h,k*). However, in animals co-injected with both pan-204 MO and nfatc1 splice MO, formation of the thoracic duct is largely restored (*Figure 7c,f,j*), as is the formation other lymphatic vessels including the DLLV and intersegmental lymphatics (ISLV) (*Figure 7c,i,k*). These results suggest that abnormal lymphatic vessel development in mir-204 deficient animals can be substantially rescued by suppressing nfatc1 expression. Based on all of our findings, we propose that a proper balance between mir-204 and nfatc1 is critical for proper lymphatic vessel development, with loss of either mir-204 or nfatc1 causing defects in lymphatic vessel formation (*Figure 7l*).

## Discussion

Although transcriptional programs directing lymphatic vessel formation have been described in recent years (*Wigle and Oliver, 1999*; *Dumont et al., 1998*; *Karkkainen et al., 2004*; *François et al., 2008*; *Srinivasan et al., 2010*; *Yuan et al., 2002*; *Uhrin et al., 2010*; *Hogan et al., 2009*; *Kulkarni et al., 2009*; *Sweet et al., 2015*; *Bui et al., 2016*; *Cho et al., 2019*), the post-transcriptional steps that help to refine this tightly regulated process remain largely unexplored. In this study, we identified LEC-enriched microRNAs by comparing the small RNA profiles of human primary blood (HUVEC) or lymphatic (HMVEC-dLy) endothelial cells. Both of these cells have been previously well-validated as having representative blood endothelial cell (BEC) and or lymphatic endothelial cell (LEC) gene expression profiles (*Leslie Pedrioli et al., 2010*; *Dunworth et al., 2014*), and we confirmed that they have appropriate differential expression of well-characterized markers of blood and lymphatic endothelial identity. The majority of the 30 highly significantly LEC-enriched microRNAs we identified were specific to mammals, with only seven microRNAs conserved in other vertebrate species. This suggests that most of these LEC microRNAs have diverged and/or evolved for mammalian-specific functions. The most highly LEC-enriched microRNA was miR-204, a microRNA with 100% sequence identity across a broad swath of vertebrate species (*Figure 2—figure supplement 2a*), suggesting it plays an important conserved role during lymphatic development. In addition to miR-204 and a number of other newly identified LEC-enriched microRNAs, we also uncovered microRNAs such as miR-326, miR-139, miR-338, miR-148a, and miR-30d that had been previously reported to be LEC-enriched (*Leslie Pedrioli et al., 2010*). Intriguingly, miR-204 was not identified in the previous study by Pedrioli et al, despite its being the most highly enriched lymphatic microRNA in our study, while the most enriched LEC microRNA reported by Pedrioli et al, miR-95, was excluded from our analysis due to a low number of sequence reads. The reasons for the disparate results obtained from our data set and that of Pedrioli et al. are not clear, but they may reflect differences in the starting material for sequencing, the profiling methods used, or the bioinformatic tools and filters employed. We would note that out study used three biological replicates for both the LEC and BEC sequencing, and that we sequenced to a relatively high depth and applied very stringent filtering to our data set. Our FACS sorting data confirmed very strong enrichment of miR-204 in zebrafish lymphatic vs. blood endothelial cells. Although we were not able to detect this microRNA in zebrafish using previously reported whole mount in situ hybridization methods (*Wienholds et al., 2005*), detection of microRNAs in zebrafish using this method is difficult and frequently unsuccessful, especially for mid- or low-copy microRNAs (*Koshiol et al., 2010*).

The remarkable evolutionary conservation of the miR-204 sequence throughout the vertebrates suggests that this microRNA likely plays an important conserved function. Indeed, our zebrafish findings show that suppression of mir-204 leads to strong defects in developmental lymphangiogenesis. Although the sequence of mature zebrafish mir-204 is identical to human miR-204, human miR-204 is encoded by only a single locus in intron 6 of the TRPM3 gene, while zebrafish mir-204 is encoded by

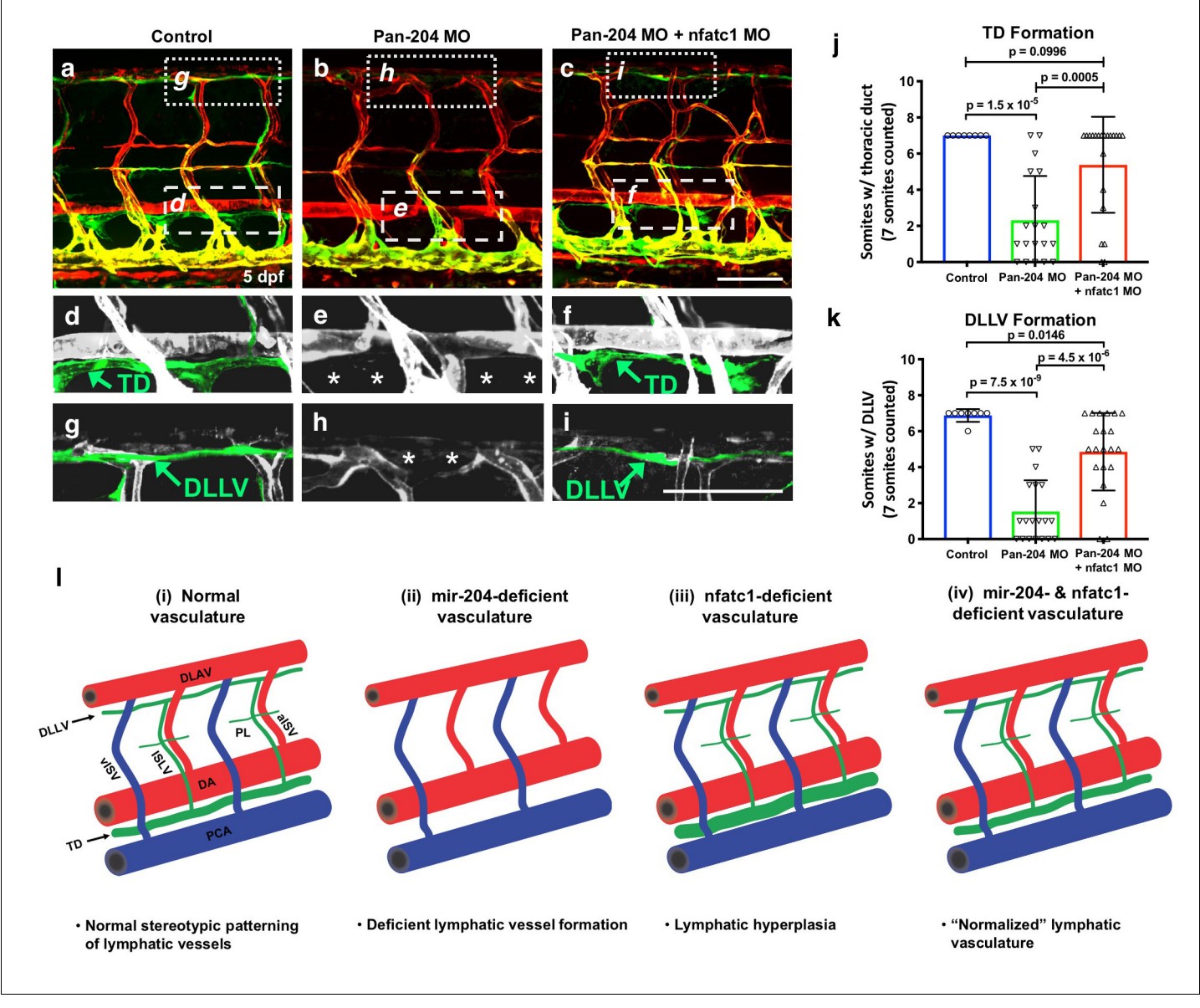

**Figure 7.** Suppression of nfatc1 rescues the lymphatic defects in mir-204-deficient animals. (**a–c**) Confocal images of the mid-trunk of 5 dpf control (**a**) pan-204 MO-injected (**b**) or pan-204 MO and nfatc1 splice MO co-injected (**c**) animals. White dotted boxes in panels a-c show areas magnified in panels d-f, respectively, while white dashed boxes show areas magnified in panels g-i, respectively. (**d–f**) Magnified images from panels a-c with the thoracic duct (TD) pseudocolored in green and other vessels in gray. The TD is labeled, and the absence of the TD is noted with asterisks. (**g–i**) Magnified images from panels a-c with the dorsal longitudinal lymphatic vessel (DLLV) pseudocolored in green and other vessels in gray. The DLLV is labeled, and the absence of the DLLV is noted with asterisks. (**j**) Quantification of thoracic duct (TD) formation in five dpf control (n = 8), pan-204 MO-injected (n = 19), or pan-204 MO and nfatc1 splice MO co-injected animals (n = 21). A total of 7 mid-trunk somitic segments were scored in each animal for the presence or absence of an intact TD. (**k**) Quantification of dorsal longitudinal lymphatic vessel (DLLV) formation in five dpf control (n = 8), pan-204 MO-injected (n = 19), or pan-204 MO and nfatc1 splice MO co-injected animals (n = 21). A total of 7 mid-trunk somitic segments were scored in each animal for the presence or absence of an intact DLLV. (**l**) Schematic diagrams illustrating five dpf zebrafish trunk lymphatic vessels present in (**i**) normal control, (**ii**) mir-204 deficient, (**iii**) nfatc1-deficient, and (**iv**) mir-204- and nfatc1-deficient animals. Suppression of mir-204 leads to loss of lymphatic vessels (ii), while nfatc1 deficiency causes lymphatic (thoracic duct) hyperplasia (iii). The lymphatic defects in mir-204 deficient animals can be rescued by simultaneous suppression of nfatc1 (iv). All images are lateral views of $Tg(mrc1a:eGFP)^{y251}$, $Tg(kdrl:mCherry)^{y171}$ double-transgenic animals, rostral to the left. Scale bar = 100 μm (**c,i**). All graphs are analyzed by t-test and the mean ± SD is shown.

DOI: https://doi.org/10.7554/eLife.46007.024

The following source data is available for figure 7:

**Source data 1.** Numerical data for *Figure 7*.
DOI: https://doi.org/10.7554/eLife.46007.025

three separate loci within the introns of the *trpm3, trpm1a,* and *trpm1b* genes. Interestingly, however, the mammalian *TRPM1* gene has a related microRNA encoded within a similar intron to zebrafish *trpm1a* and *trpm1b,* that may have evolved from mir-204. We demonstrated the essential role of miR-204 in developmental lymphangiogenesis using multiple complementary loss-of-function approaches, including (i) a 'pan-204' morpholino targeting the mature mir-204 sequence produced by all three zebrafish mir-204 loci, (ii) morpholinos individually targeting sequences required for the maturation of each of the three zebrafish mir-204's (MO1, MO2, and MO3), and (iii) a CRISPR-Cas9-generated genetic mutant ablating the *mir-204–1* locus, which caused a lymphatic defect in combination with MO2 (targeting miR-204–2). Our results revealed that treatments that target at minimum *mir-204–1* and *mir-204–2* result in comparable dramatic loss-of-lymphatic phenotypes, including loss of the parachordal line at three dpf and loss of the thoracic duct and other trunk lymphatic vessels at five dpf.

Our endothelial-specific transgenic miR-204 expression experiments confirm the important role of miR-204 in lymphatic development in the zebrafish, and further indicate that this role reflects endothelial-autonomous function of this microRNA. Wild type animals injected with mrc1a:mir204-1-eGFP transgene display precocious thoracic duct development, as do germline transgenic animals carrying the transgene inserted into the genome. Importantly, the mrc1a:mir204-1-eGFP transgene also 'rescues' thoracic duct formation when introduced into miR-204–1$^{-/-}$ mutants injected with MO2 (targeting miR-204–2) that would otherwise lack lymphatics, demonstrating that re-introduction of miR-204 exclusively in the endothelium is sufficient to restore normal lymphatic development in miR-204-deficient animals. Despite all of these interlocking findings strongly arguing for a key role for miR-204 in lymphatic development, triple mutants disrupting all three zebrafish miR-204 loci do not display a significantly measurable lymphatic defect. Although the reasons for this are unclear, there is precedent in the literature for zebrafish microRNA mutants failing to exhibit the full phenotypes noted in microRNA knockdown studies. Suppression of the evolutionarily conserved endothelial-enriched microRNA mir-126 causes blood vessels defects in mir-126 morpholino-injected animals (*Fish et al., 2008*) (*Zou et al., 2011*), although lower dose morpholino injections cause mainly lymphatic defects with relatively minor effects on blood vessels (*Chen et al., 2016*). Interestingly, mir-126 mutant zebrafish also lack the vascular phenotype described in the previous morpholino studies and display only the defects in lymphatic vessel development seen with partial knockdowns (*Kontarakis et al., 2018*), suggesting compensation may be taking place in miR-126 mutants.

In addition to uncovering an important function for miR-204, our study also identified a key downstream target of miR-204 regulation during lymphangiogenesis. Previous studies have shown that the Nuclear Factor of Activated T-cells (NFAT) protein family member NFATC1 is important for lymphatic vessel development (*Norrmén et al., 2009*; *Kulkarni et al., 2009*). Nfatc1 is co-expressed with Prox1-, Vegfr3-, and Pdpn-positive LEC progenitors originating from the cardinal vein indicating their potential involvement in lymphatic specification, and Nfatc1 null mice develop lymphatic hyperplasia suggesting a role in lymphatic maturation (*Norrmén et al., 2009*; *Kulkarni et al., 2009*). Consistent with these previous data, we showed that nfatc1 mutation, nfatc1 knockdown, or pharmacologically blocking nfatc1 activity results in similar lymphatic enlargement in the zebrafish. We identified NFATC1 as a probable miR-204 target based on conserved miR-204 binding sites in the 3' UTRs of both human and zebrafish NFATC1 transcripts and the expression of NFATC1 was suppressed by miR-204 mimic and increased by miR-204 antagomir in human LECs. We were able to verify targeting of the NFATC1 3' UTR by miR-204 using firefly/renilla dual luciferase NFATC1 3' UTR reporter assays, and demonstrate up-regulation of zebrafish nfatc1 upon mir-204 knockdown, confirming that miR-204 suppresses NFATC1 levels in vivo. Finally, we showed that knocking down nfatc1 could rescue lymphatic development in mir-204-deficient zebrafish, suggesting that balanced expression of mir-204 and nfatc1 is critical for proper developmental lymphangiogenesis.

Since microRNAs often regulate many targets, and miR-204 does have potential target binding sites in other genes, further investigation will be required to characterize some of these additional genes and determine whether some of the lymphatic effects of miR-204 are mediated via regulation of other targets in addition to nfatc1. Nevertheless, since (i) NFATC1 has already been shown to play an important role in lymphangiogenesis in mammals, (ii) suppression of nfatc1 in the zebrafish causes lymphatic hyperplasia, and (iii) nfatc1 knockdown effectively rescues the effects of miR-204 knockdown, our results suggest that nfatc1 is a major, key downstream target regulated by miR-204 in developing lymphatic endothelial cells.

In summary, our work establishes an important role for miR-204 in regulating lymphatic vascular network formation during embryonic development via modulation of its conserved target NFATC1. Our findings provide important new insight into the role of lymphatic-enriched microRNAs during developmental lymphangiogenesis, and by unveiling microRNA-regulated pathways provide new opportunities to understand lymphatic development and associated disorders.

# Materials and methods

## Key resources table

| Reagent type (species) or resource | Designation | Source or reference | Identifiers | Additional information |
|---|---|---|---|---|
| Genetic reagent (*D. rerio*) | Tg(mrc1a:eGFP$^{y251}$; kdrl:mCherry$^{y171}$) | PMID: 28506987 | ZFIN ID: ZDB-TGCONSTRCT-170717–2 | |
| Genetic reagent (*D. rerio*) | Tg(mrc1a:eGFP$^{y251}$) | PMID: 28506987 | ZFIN ID: ZDB-TGCONSTRCT-170717–2 | |
| Genetic reagent (*D. rerio*) | Tg(mrc1a:mir 204-eGFP) | This paper | | |
| Genetic reagent (*D. rerio*) | mir-204–1$^{-/-}$; mir-204–2$^{-/-}$; mir-204–3$^{-/-}$ | This paper | | |
| Genetic reagent (*D. rerio*) | nfatc1$^{\triangle8/\triangle8}$ | This paper | | |
| Cell line (*H. sapiens*) | HMVEC-dLy | Lonza | Cat# CC-2812 | |
| Cell line (*H. sapiens*) | HUVEC | GIBCO | Cat# C-003–5C | |
| Cell line (*H. sapiens*) | HEK293 | ATCC | Cat# CRL-1573 RRID:CVCL_0045 | |
| Recombinant DNA | pME-mir204 | This paper | | Used pME-miR for backbone vector (PMID: 2914488) |
| Recombinant DNA | mrc1a:mir204-eGFP | This paper | | |
| Recombinant DNA | mrc1a:mir204 (4 bp_seed_mut)-eGFP | This paper | | |
| Recombinant DNA | psiCheck2-hNFATC1-3′UTR | This paper | | Used for luciferase assay |
| Recombinant DNA | psiCheck2-hNFATC1-3′UTR_4 bp_mut | This paper | | Used for luciferase assay |
| Recombinant DNA | psiCheck2-zNFATC1-3′UTR | This paper | | Used for luciferase assay |
| Recombinant DNA | psiCheck2-zNFATC1-3′UTR_4 bp_mut | This paper | | Used for luciferase assay |

*Continued on next page*

*Continued*

| Reagent type (species) or resource | Designation | Source or reference | Identifiers | Additional information |
|---|---|---|---|---|
| Sequence-based reagent | Cloning primers | This paper | | See *Supplementary file 2a* for sequence information |
| Sequence-based reagent | Morpholinos | This paper | | See *Supplementary file 2b* for sequence information |
| Sequence-based reagent | gRNAs | This paper | | See *Supplementary file 2c* for sequence information |
| Sequence-based reagent | hsa-miR-204–5 p mimic | Thermo Fisher Scientific | Cat# MC11116 | |
| Sequence-based reagent | hsa-miR-204–5 p inhibitor | Thermo Fisher Scientific | Cat# MH11116 | |
| Sequence-based reagent | hsa-miR-126–3 p mimic | Thermo Fisher Scientific | Cat# MC12841 | |
| Chemical compound, drug | Cyclosporine A | Sigma-Aldrich | Cat# PHR1092 | |
| Septide, recombinant protein | EnGen Lba Cas12a (Cpf1) | New England Biolabs | Cat# M0653T | |
| Software, algorithm | ImageJ | ImageJ (http://imagej.nih.gov/ij/) | RRID:SCR_003070 | |
| Software, algorithm | Imaris | Bitplane | RRID:SCR_007370 | |
| Software, algorithm | Adobe Photoshop | Adobe | RRID:SCR_014199 | |
| Software, algorithm | NIS-Elements | Nikon | RRID:SCR_014329 | |
| Software, algorithm | GraphPad Prism | GraphPad Prism (https://graphpad.com) | RRID:SCR_015807 | |

## Zebrafish and drug treatment

Zebrafish husbandry and research protocols were reviewed and approved by the NICHD Animal Care and Use Committee at the National Institutes of Health (Animal Research Assurance Number: A4149-01). All animal studies were carried out according to NIH-approved protocols (Animal Study Proposal: #18–015), in compliance with the *Guide for the Care and use of Laboratory Animals*. Zebrafish were maintained and zebrafish experiments were performed according to standard protocols (*Westerfield, 2000*). The *Tg(mrc1a:eGFP)^{y251};Tg(kdrl:mCherry)^{y171}* double transgenic line was used in this study (*Jung et al., 2017*). For nfatc1 inhibition, 24 hpf embryos were dechorinated and incubated in 1 ug/mL cyclosporine A (CsA) or DMSO for 4 days and the animals imaged at five dpf.

## Transgenic constructs and animals

The *Tol2(mrc1a:mir204-eGFP)* venous/lymphatic endothelial autonomous expression construct was generated using Tol2kit components with Gateway Technology (*Kwan et al., 2007*). To make a *mir-204* middle entry cassette for the overexpression construct, we used pME-miR, containing an partial EF1alpha gene exon 1, intron 1, and exon 2 (*Nicoli et al., 2010*). A 1 kb genomic DNA sequence from intron 5 of the *trpm3* gene harboring the *mir-204–1* precursor was amplified by PCR and subcloned into the multiple cloning site located in EF1alpha intron 1 of pME-miR using Kpn1 and EcoR1, to generate pME-mir204. To generate the final venous/lymphatic endothelial autonomous mir-204 expression construct, we combined p5E-mrc1a (*Jung et al., 2017*), pME-mir204, p3E-EGFPpA (*Kwan et al., 2007*), and pDestTol2pA (*Kwan et al., 2007*). Mutations in the miR-204 seed sequence were generated using the QuikChange site-directed mutagenesis kit (Stratagene, La Jolla, CA) and

the sequences were confirmed by DNA sequencing. The DNA construct was microinjected into one-cell stage zebrafish embryos to generate transgenic insertions. Injected animals were either analyzed during early development or raised to adulthood and their progeny screened for germline transmission and expression of the transgene. All oligos used for cloning are listed (*Supplementary file 2a*).

## Flow cytometry

All embryos subjected to FACS sorting were raised in E3 medium. five dpf *Tg(mrc1a:egfp)^{y251}*, *Tg(kdrl:mcherry)^{y171}* double transgenic zebrafish were anesthetized with MS-222 and washed with 1X PBS (pH 7.4, without $Ca^{2+}$ and $Mg^{2+}$) three times. Animals were deyolked by gentle pipetting in yolk dissociation solution (55 mM NaCl, 1.8 mM KCl, 1.25 mM $NaHCO_3$). Cells were then dissociated by gentle pipetting in 0.25% trypsin-EDTA and 50 mg/mL collagenase solution. Dissociated cells were passed though 70 µm filter and centrifuged at 4000 rpm for 5 min at room temperature. Cells were washed and resuspended with 1X PBS. Fluorescent cell sorting was performed on a BD FACS ARIA (Becton Dickinson, Franklin Lakes, NJ). Isolated GFP+, mCherry+, and double positive cells were pelleted at 2,500 rpm for 5 min.

## Morpholino injections

All morpholinos (MOs) used in this study were acquired from Gene Tools. The nfatc1 splice MO sequence was described in a previous study (*Tijssen et al., 2011*). MOs were injected into one cell stage *Tg(mrc1a:egfp)^{y251}*, *Tg(kdrl:mcherry)^{y171}* double transgenic zebrafish embryos (*Jung et al., 2017*). Injected embryos were allowed to develop at 28.5°C before being imaged at the desired stage. Morpholino doses used were determined by performing dose curves to establish the optimal dose to minimize off-target effects. The doses used in this study were 0.5 ng for all mir-204 MOs (pan-204, MO1, MO2, and MO3), 3 ng for nfatc1 splice MO, and 7.5 ng for nfatc1 ATG MO. All morpholino sequences are listed (*Supplementary file 2b*).

## Genome editing

CRISPR genome editing technology was used to generated miR-204 mutants. In order to use the most efficient editing by finding the optimal protospacer adjacent motif (PAM) sequence, Cas9 was used to target *dre-miR-204–1* and Cpf1 (Cas12a) was used to target *dre-miR-204–2* and *dre-miR-204–3*. The codon-optimized Cas9 plasmid pT3TS-nls-zCas9-nls was used as template to in vitro transcribe Cas9 mRNA (*Jao et al., 2013*) and we used commercially available Cpf1 protein (New England Biolabs) as described (*Fernandez et al., 2018*; *Moreno-Mateos et al., 2017*). Single guide RNAs (sgRNAs) were designed to target each precursor *mir-204* sequence using CRISPRscan (www.crisprscan.org) (*Moreno-Mateos et al., 2015*). The sgRNA for *mir-204–1* was in vitro synthesized using T7 promoter, and sgRNAs for *mir-204–2* and *mir-204–3* were obtained from Integrated DNA Technologies (IDT). Fluorescence PCR was performed using AmpliTaq Gold DNA polymerase (Life Technologies) with M13F primer with fluorescence tag (6-FAM), amplicon-specific forward primer with M13 forward tail (5'-TGTAAAACGACGGCCAGT-3') and 5'PIG-tailed (5'-GTGTCTT-3') amplicon-specific reverse primer for genotyping to identify alleles that contain a large deletion on the seed sequence of mir-204 (*Varshney et al., 2015*). An ABI 3730 Genetic Analyzer Avant (Thermofisher) was used to analyze the PCR products. All oligos used for genome editing are listed (*Supplementary file 2c*).

## Cell culture and transfection

HMVEC-dLy (Lonza), HUVEC (GIBCO), and HEK293 (ATCC) were purchased and no evidence of *Mycoplasma* contamination was found. HMVEC-dLy cells (Lonza) were cultured in EGM-2 MV Bullet-Kit (Lonza, CC-3202) that contains hEGF, hydrocortisone, GA-1000, FBS, VEGF, hFGF-B, $R^3$-IGF-1, and ascorbic acid. HUVECs (Lonza) were cultured in bovine hypothalamus extract, 0.01% Heparin and 20% FBS in M199 base media (Gibco) on 1 mg/mL gelatin-coated tissue culture flasks. HEK293 cells were cultured in Advanced DMEM supplemented with 10% FBS and antibiotics. Transfection was performed using Lipofectamine 2000 (Invitrogen).

## RNA isolation, small RNA-seq, and TaqMan PCR

RNA isolation was performed using the mirVana kit (Life Technoloiges). A NanoDrop ND-100 spectrophotometer (Nanodrop Technology Inc), Qubit 2.0 fluorometer (Life Technologies Inc) and Agilent 2100 bioanalyzer (Aglient) were used to analyze RNA quantity and quality. Small RNA sequencing was performed by ACGT, Inc. Three biological samples were subjected to analysis. Briefly, libraries were prepared using the Illumina TruSeq Small RNA Sample Preparation Kit, then sequenced on a NextSeq 500 Illumina instrument, generating 50 bp single end reads. Data was analyzed using PartekFlow analysis software (Partek, Inc). The sequence reads were trimmed to remove the following adapters: GTTCAGAGTTCTACAGTCCGACGATC (from the 5' end) and TGGAATTC TCGGGTGCCAAGG (from the 3' end). Then, the bases at the end of the sequences with quality less than 20 were removed. The remaining sequences were aligned to the human genome browser (hg38) and miRbase mature microRNA version 22 using Bowtie. The data was filtered for the counts smaller than 10 in 50% of samples, and used CPM (counts per million) for normalization. Differential expression was analyzed by Partek GSA algorithm. All supplies for TaqMan microRNA/gene assays were purchased from Life Technologies, and qPCR was performed using a CFX96 (BioRad). The mature microRNA sequences of all three zebrafish mir-204s are identical and could therefore be detected using a common mir-204 TaqMan assay. 18S rRNA (for human cells) and ef1a (for zebrafish cells) were used for internal controls for mRNAs, and U6 snRNA or mir-126 was used as an internal control for microRNAs.

## Luciferase reporter assay

The human NFATC1 3'UTR was PCR amplified from cDNA generated from human LEC RNA, and the zebrafish nfatc1 3'UTR from zebrafish embryo RNA. 3'UTR sequences were cloned downstream from the *renilla* luciferase gene using the XhoI and NotI sites in the psiCHECK-2 vector (Promega, Madison, WI). This vector also contains a *firefly* luciferase gene driven by an independent protomer, which serves an internal control for the assay. Mutations in the miR-204 binding sites were generated using the QuikChange site-directed mutagenesis kit (Stratagene, La Jolla, CA) and the mutated sequences were confirmed by DNA sequencing. All the primers used for generating luciferase constructs are listed (*Supplementary file 2a*). HEK293 cells transfected with luciferase reporters and microRNA mimics were harvested after 24 hr. A dual luciferase reporter assay system was used to determine luciferase levels (Promega, Madison, WI).

## Imaging methods

Embryos were anesthetized using 1x tricaine and mounted in 0.8–1.5% low melting point agarose dissolved in embryos media and mounted on a depression slide (*Jung et al., 2016*). Time-lapse imaging was performed ~20 hr with image stacks acquired every 8 min. Confocal fluorescence imaging was performed with a Nikon Yokogawa CSU-W1 spinning disk confocal microscope. The images were analyzed using ImageJ (*Schindelin et al., 2012*), Imaris 7.4 (Bitplane), Adobe Photoshop (Adobe), and NIS-Elements (Nikon) software.

## Statistical analysis

Statistical significance was determined by using Student's t-test as indicated in the corresponding figure legends. At least three biological replicates per condition were used for quantitation. A biological replicate is defined as individual embryos or an independent culture of cells. The number of replicates are indicated in the figures or the legends. All quantitative data was analyzed used GraphPad Prism and Student's t-test.

Conflict-of-interest disclosure: The authors declare no competing financial interests.

# Acknowledgements

The authors would like to thank members of the Weinstein laboratory for their critical comments on this manuscript. This work was supported by the intramural program of the *Eunice Kennedy Shriver* National Institute of Child Health and Human Development, National Institutes of Health (ZIA-HD008808 and ZIA-HD001011, to BMW).

# Additional information

### Funding

| Funder | Grant reference number | Author |
|---|---|---|
| National Institutes of Health | ZIA-HD008808 | Brant M Weinstein |
| National Institutes of Health | ZIA-HD001011 | Brant M Weinstein |

The funders had no role in study design, data collection and interpretation, or the decision to submit the work for publication.

### Author contributions

Hyun Min Jung, Conceptualization, Data curation, Formal analysis, Validation, Investigation, Visualization, Methodology, Writing—original draft, Writing—review and editing; Ciara T Hu, Alexandra M Fister, Data curation, Formal analysis, Investigation; Andrew E Davis, Van N Pham, Data curation, Investigation; Daniel Castranova, Data curation, Formal analysis; Lisa M Price, Data curation; Brant M Weinstein, Conceptualization, Supervision, Funding acquisition, Validation, Writing—review and editing

### Author ORCIDs

Hyun Min Jung (iD) https://orcid.org/0000-0001-8892-0941
Brant M Weinstein (iD) https://orcid.org/0000-0003-4136-4962

### Ethics

Animal experimentation: Zebrafish husbandry and research protocols were reviewed and approved by the NICHD Animal Care and Use Committee at the National Institutes of Health (Animal Research Assurance Number: A4149-01). All animal studies were carried out according to NIH-approved protocols (Animal Study Proposal: #18-015), in compliance with the Guide for the Care and use of Laboratory Animals. Zebrafish were maintained and zebrafish experiments were performed according to standard protocols (Westerfield, 2000).

### Decision letter and Author response

Decision letter https://doi.org/10.7554/eLife.46007.032
Author response https://doi.org/10.7554/eLife.46007.033

# Additional files

### Supplementary files

• Supplementary file 1. List of lymphatic genes analyzed as putative miR-204 targets in both human and zebrafish.
DOI: https://doi.org/10.7554/eLife.46007.026

• Supplementary file 2. List of sequence information for the primers, morpholinos, and oligonucleotides used for CRISPR genome editing.
DOI: https://doi.org/10.7554/eLife.46007.027

• Transparent reporting form
DOI: https://doi.org/10.7554/eLife.46007.028

### Data availability

All data generated or analyzed during this study are included in the manuscript and supporting files. Source data files have been provided for all the figures. Sequencing data have been deposited in GEO under accession codes GSE126679.

The following dataset was generated:

| Author(s) | Year | Dataset title | Dataset URL | Database and Identifier |
|---|---|---|---|---|
| Hyun Min Jung, Brant Weinstein | 2019 | MicroRNA expression in human lymphatic endothelial cells (LEC) and blood endothelial cells (BEC) | https://www.ncbi.nlm.nih.gov/geo/query/acc.cgi?acc=GSE126679 | NCBI Gene Expression Omnibus, GSE126679 |

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
