## [Decision Letter]

Thank you for sending your article entitled "MicroRNA-mediated control of developmental lymphangiogenesis" for peer review at *eLife*. Your article is being evaluated by three peer reviewers, one of whom is a member of our Board of Reviewing Editors, and the evaluation has been overseen by Marianne Bronner as the Senior Editor.

In the light of the comments made by the reviewers, we feel that significant revisions will be necessary before your manuscript can be published in *eLife*.

The comments of all three reviewers are in good agreement. While the reviewers found this work to be of some interest, they raised concerns about the strength of the conclusions that can be drawn at this stage and the appropriateness of the technical approach. The authors would be required to carefully address all comments point-by-point in a data-driven manner or with further analyses. Specifically, more expression analyses and better genetic evidence are required. Given that the authors intend to propose a link between miR-204 and Nfatc1, a more definite mechanism study that is supported by additional experiments is required. A stable Nfatc1 mutant instead of a morphant or CsA treatment would be a key control that is also essential. Double-knockout of mir-204-2 and mir-204-3 is also required to demonstrate the indispensable roles of mir-204. Please ensure your plan addresses these concerns and, if necessary, please provide the reasons for not implementing the suggested changes.

*Reviewer #1:*

The post-transcriptional mechanisms in lymphatic vessel formation and patterning are barely understood. This is because it is still challenging to analyze phenotypes and investigate molecular mechanisms in miRNA-deficient model organisms. In this study, the authors successfully uncovered lymphatic-enriched miRNAs by small RNA sequencing analysis. In addition, they found interesting lymphatic phenotypes in the miR-204-deficient zebrafish and unveiled a target transcriptional factor regulated by miR-204, NFATC1. With this data and their pipeline for validating the functionality of lymphatic-enriched miRNAs, this study is expected to contribute to follow-up studies to determine post-transcriptional mechanisms in lymphatic vessel formation.

However, the role of miR-204 in lymphatic vessel formation is not yet conclusive and requires further clarification and investigation. It is still uncertain whether miR-204/NFATC1 interaction contributes to lymphatic vessel formation or patterning. Furthermore, how the expression of miR-204 is regulated during lymphatic vessel development needs to be investigated.

1) In Figure 1, how is the expression of miR-204 being regulated during lymphatic vessel development? Is miR-204 always highly expressed in developing lymphatic endothelial cells (i.e. zebrafish lymphatic endothelial cells at 5dpf) and mature lymphatic endothelial cells (i.e. HMVEC-dLy), or is it dynamically regulated in a time-dependent manner?

2) In Figure 2C, what is the authors' opinion on the cause of different phenotypes depending on the dose of morpholinos?

3) It is still uncertain whether miR-204/NFATC1 molecular pathway contributes to lymphatic vessel formation or patterning. In Figure 2E-N, Figure 3 and Figure 4, authors showed that miR-204 contributes to lymphatic vessel formation by loss-of and gain-of-function experiments. On the other hand, in Figure 6, nfatc1 seems to be required for lymphatic vessel patterning, not lymphatic vessel formation. According to the references the authors provided (Norrmen et al., 2009, Kulkarni et al., 2009), nfatc1 in mice also plays roles in lymphatic vessel patterning, not lymphatic vessel formation. Overall, miR-204 seems to work not only with nfatc1 signaling pathway but also with other signaling pathways, especially in lymphatic vessel formation. The authors need to clarify this issue.

4) In line with comment 3, in Figure 7, it is unclear how deficient 'lymphatic vessel formation' by Pan-204 MO could be rescued by 'lymphatic enlargement' induced by nfatc1 MO. Careful analysis is required.

*Reviewer #2:*

Jung et al. identify miR-204 as highly expressed in cultured human LECs and in sorted populations of early developing LECs from the zebrafish. They go on to show that morpholino knockdown of the 3 different mir-204 transcripts (from 3 genes) can lead to a loss of lymphatic vessel development. Importantly, they generate a miR-204-1 mutant animal and show that some combinations of MOs injected into this mutant can lead to a phenotype specific to the mutant embryos. They go onto identify NFATC1 as a possible target of miR-204 and to suggest a mechanism whereby miR-204 functions to suppress NFATC1 expression and that NFATC1 is itself a negative regulator of lymphatic vessel size.

While the work is interesting and may offer up an unexpected new mechanism controlling lymphatic development. There are several questions remaining and loose ends that should be better completed to give more confidence in the central findings.

Major issues:

1) miR-204 has been previously analysed in zebrafish and medaka and prominent expression reported in epithelia and the eye (retinal epithelia and lens). In these previous studies using LNA in situ hybridisation (Conte et al., 2010, Weinholds et al., 2005) there was no indication of vascular expression. The authors should provide evidence with a similar approach that miR-204 is expressed autonomously in intact embryos in the vasculature.

It is possible that the function of miR-204 may be non-autonomous in the current report and so at least showing in situ expression in tissues would improve confidence in the current data.

2) The paper relies heavily on the use of MO knockdown which has become increasingly controversial in the zebrafish field. Having one mutant in miR-204-1 is welcome and does improve confidence. However, while this reviewer appreciates that asking for triple mutants is perhaps too much, there are some inconsistencies that are concerning. For example, MO1 + MO2 gives a loss of lymphatics but the mutant for 204-1 + MO2 does not. How can this be explained? Can additional evidence such as transient CRISPR for 204-2 or -3 in the miR-204-1 mutant or similar be provided to further improve confidence in these data with multiple overlapping approaches?

3). The analysis of phenotype is very superficial. Is specification of LECs impacted? Is LEC cell number at the parachordal line or thoracic duct quantitatively reduced? Is there signalling induced downstream of Vegf-c and Vegf-r3? Eg. pERK such as shown in Shin et al., 2016.

Along the same lines the phenotypic analysis of the overexpression transgenic for miR-204 and in the NFATC1 vessels should include cell number counts for LECs. This is important as scoring vessel area in 2D images could indicate increased LECs or increased vessel dilation (more luminal content). The NFATC1 mutant could be a fluid imbalance phenotype and unrelated to the miR-204 overexpression phenotype or miR-204 mutant/MO phenotype without further more careful phenotypic analysis.

4). The overexpression of miR-204 gives premature thoracic duct development but the embryo shown looks older than the control (increased distance between DA and PCV). The experiment is also under-controlled. Please provide analysis of markers of other tissues to control for staging differences (eg. rag1 expression in thymus is a useful marker that comes on progressively from around 2.5 dpf). Please also show how much the endothelial levels of miR-204 are increased in the transgenic – miRNA's are typically highly expressed and Figure 1 suggests very high levels of miR-204 already in LECs, so why would one expect such a phenotype upon over-expression?

5) Is NFATC1 expressed in LECs and the PCV in zebrafish by ISH? Is the expression increased upon progressive loss of miR-204 in the mutants and MO scenarios?

6) The analysis of NFATC1 phenotype to correlate with miR-204 phenotypes relies exclusively upon MO knockdown and the use of CsA, an inhibitor commonly considered to have broad impact on embryos. Unlike the miR targeting, where it is difficult to expect generation of triple mutants, analysis of a stable NFATC1 mutant does not seem an unreasonable thing to ask for. It is likely that a mutant may be available already. The authors should provide more confidence in their increased lymphangiogenesis phenotype upon NFATC1 loss of function by including the analysis of a mutant strain. Alternatively (or additionally), they could consider targeting the miR-binding site in the 3'UTR to definitively demonstrate their mechanism.

7) The honing in on NFATC1 as a target came without explanation of how many miR-204 targets are predicted in vasculature. Providing additional bioinformatic prediction would improve the study. How many LEC transcripts have predicted miR-204 target sites? Is this statistically enriched over other cell types? How does NFATC1 rank as a predicted target taking into account the number of predicted sites, homology etc?

*Reviewer #3:*

Jung HM, et al. report a potential mir-204-dependent regulation of lymphatic vessel (LV) growth through NFATC1. First, the authors identified mir-204 as a lymphatic endothelial cells (LECs)-enriched miRNA compared to vascular endothelial cells (VECs). There were three mir-204 in zebrafish: mir-204-1, (in the intron of 5 of trpm3), mir-204-2 (in the intron of trp1ma), and mir-204-3 (in the intron of 4 of trp1mb). They demonstrated the requirement of miR-204 in lymphatic vessel development by using mir-204-1 mutants treated with both mir-204-2 and -3 morpholinos and mir-204-1, -2, -3 morphants (total morphants). Furthermore, NFATC1 was reported to be a potential target of mir-204, because the transcription of nfatc1 was increased in the mir-204 total morphants and was decreased in the reporter assay using mir-204. They also demonstrated the potential involvement of NFATC1 in LV growth by showing nfatc1 morphants and a calcineurin inhibitor, cyclosporin A. The hierarchical gene regulation between mir-204 and nfatc1 for LV development was validated by the restoration of LV impairment found in mir-204 morphants in the mir-204 morphants treated with nfatc1 morpholinos.

This study is well organized and written in a logical manner. There are still several points that should be addressed to support their claim.

1) It is unclear why the authors need to compare zebrafish and human? The conservation of mir-204-dependent regulation of NFATC1 among zebrafish and human might be interesting: however, the authors could identify mir-201 using mir-RNAs from the *Tg(mrc1a:eGFP)^y251^* and *Tg(kdrl:mCherry)^y171^* as shown in Figure 1F. Because the importance of mir-204 in zebrafish is focused in this study, the list of mir using the mir from the FACS-sorted cells should be used as an initial material. The authors might do this experiment easily using their Tg fish.

The result shown in Figure 1A, B, C should become Figure supplement.

2) In addition to #1, time course of mir expression should be demonstrated at least at two points when the secondary sprout starts and when thoracic duct formation is completed. The authors merely showed the expression of mir-204 in the zebrafish embryos at 5 dpf.

3) The requirement of mir-204 is partially proven in this study. As the authors explained in Figure 3, they need to do the similar experiments using mir-204-^2 -/-^ + MO1, 3 and mir-204-3 ^-/-^ + MO1, 2. It is not so difficult to complete these experiments by establishing mir-204-2 mutant and mir-204-3 mutant. They describe the detail of mir-204-2 and mir-204-3 in Figure 2—figure supplement 1.

4) The potential target of mir-204 is not described well. The authors described this in the Discussion. The reason why Nfatc1 was picked up is not clearly described in the present manuscript. The authors examined Nfatc1 following the prediction of RNA22 database. There are several microRNA target databases beside RNA22. Thus, targets expected in other database should be examined as a negative control.

5) To more precisely show the hierarchical regulation of mir-204 and nfatc1, the authors can examine the expression of nfatc1 using FACS-sorted LECs of *Tg8mrc1a:eGFP)^y251^* total morphants (mir-204-1, -2,-3 morphants).

6) It is unclear whether secondary sprout is affected in mir-204 deficient embryos. Although there are no parachordal lines in the scheme of Figure 7L, there is no evidence of impairment of secondary sprout followed by parachordal line development in the present study. Although the authors focused on thoracic duct formation, prior to this, secondary sprout and parachordal lines could be analyzed. No description in Figure 3.

7) The authors mentioned "lymphatic specification" in Discussion. They can interpret their data when carefully examine the effect of depletion of mir-204 and nfatc1 on secondary sprout and subsequent parachordal lines.

---

## [Author Response]

The comments of all three reviewers are in good agreement. While the reviewers found this work to be of some interest, they raised concerns about the strength of the conclusions that can be drawn at this stage and the appropriateness of the technical approach. The authors would be required to carefully address all comments point-by-point in a data-driven manner or with further analyses. Specifically, more expression analyses and better genetic evidence are required. Given that the authors intend to propose a link between miR-204 and Nfatc1, a more definite mechanism study that is supported by additional experiments is required. A stable Nfatc1 mutant instead of a morphant or CsA treatment would be a key control that is also essential. Double-knockout of mir-204-2 and mir-204-3 is also required to demonstrate the indispensable roles of mir-204. Please ensure your plan addresses these concerns and, if necessary, please provide the reasons for not implementing the suggested changes.

In response to the reviews of our original submission we agreed on an “Action Plan” for carrying out a large number of new experiments for our revision. We completed all of the proposed experiments, yielding important new data that has further strengthened our conclusions:

Knock down miR-204-2 in the miR-204-1 mutant

As promised, we injected miR-204-2 or miR-204-3 morpholinos into miR-204-1^-/-^ mutants (Figure 3) and obtained essentially identical results to the comparable experiments we performed using a morpholino targeting miR-204-1 instead of a miR-204-1^-/-^ genetic mutant (Figure 2—figure supplement 2). That is, combined targeting of miR-204-1 and miR-204-2 gives a lymphatic phenotype, but combined targeting of miR-204-1 and miR-204-3 does not, regardless of whether a mutant or a morpholino is used to target miR204-1. The symmetry of these results again supports a specific role for miR-204 in lymphatic development. Perform miR-204 transplant/transgene mosaic knockdown and “rescue” experiments to validate and examine their endothelial- cell specific phenotypes

As promised, we carried out experiments demonstrating that endothelial-specific expression of miR-2041 “rescues” the lymphatic defects in miR-204-1^-/-^ mutants injected with a morpholino blocking maturation of miR-204-2 (Figure 4E-K). These new data strongly confirm that the lymphatic phenotype of miR-204deficient animals is due to loss of endothelial miR-204 function, and that miR-204 is required autonomously in the endothelium for generating lymphatics. We also now show that endothelial-specific mosaic expression of miR-204-1 in wild-type animals causes precocious thoracic duct development (Figure 4A-D), further confirming that this phenotype can manifest itself in an endothelial-autonomous manner in otherwise normally developing animals.

Examine early stages of lymphatic development in miR-204-deficient animals

As further promised, we carried out time-lapse imaging of “secondary sprouts” (early pre-lymphatic sprouts that emerge from the cardinal vein) to examine whether initial specification and sprouting is defective, or later patterning and growth. Our results show that secondary sprouts in miR-204 deficient animals emerge and grow up to the level of the horizontal myoseptum normally. However, while secondary sprouts in control animals stop at the myoseptum, turn laterally, and form the parachordal lines (Figure 2C), secondary sprouts in miR-204 deficient animals fail to stop at the level of the myoseptum but continue growing dorsally, in many cases eventually contributing to the venous circulation (Figure 2D, Figure 2—video 1, and Figure 3—figure supplement 1). We would note that we have not observed this unusual and unique phenotype in any of our previous studies where we functionally manipulated genes important for lymphatic development in the zebrafish.

Analyze NFATC mutants

As also promised, we generated and analyzed CRISPR mutants in *nfatc1*, including a frameshift mutation creating a premature stop codon (Figure 6F). Our new *nfatc1*^△^*^8/^*^△^*^8^*mutants exhibit the same enlarged thoracic duct phenotype (Figure 6G-K) observed in animals injected with nfatc1 splice-blocking morpholinos (Figure 6A-E). Furthermore, we have also analyzed the phenotypes of animals injected with a separate nfatc1 translation-blocking (ATG) morpholino, and this second morpholino also gives the same enlarged thoracic duct phenotype observed in nfatc1 splice-blocking morphants (Figure 6—figure supplement 1A-E). Thus, the new nfatc1 mutant and nfatc1 ATG morphants phenocopy the lymphatic phenotypes we noted previously in animals injected with splice blocking morpholino (Figure 6) or treated with cyclosporine A (Figure 6—figure supplement 1F-J).

Explain our focus on NFATC as a key miR-204 target

As also promised, we have added an explanation to our Results section detailing the reasons for our focus on NFATC as a key miR-204 target. As we write:

“Based on our observation that miR-204 plays role during lymphatic development, we began our search for potential important target genes by (i) starting with a list of human genes previously implicated in lymphatic development, then (ii) bioinformatically identifying which genes in this set had potential miR204 target sites using TargetScan, and then (iii) bioinformatically identifying which of these genes also had corresponding zebrafish orthologs with potential miR-204 target sites (in order to ensure that we were looking at key, conserved targets that we could functionally study in both human cell culture and in zebrafish (Supplementary file 1). The Nuclear Factor of Activated T Cells 1 (NFATC1) gene immediately came to our attention as a strong candidate.”

As we noted previously, we recognize of course that microRNAs can have hundreds of potential targets, and our original goal in testing nfatc1 had been mainly to validate that miR-204 did indeed target and regulate at least one gene important for lymphatic development. It was a bit of a surprise to us that nfatc1 knockdown (or mutants) gave us a nice lymphatic phenotype, and a big (but very gratifying!) surprise that we could so effectively rescue miR-204 deficiency by knocking down nfatc1.

Analyze mir-204 mutants

As promised, we generated and analyzed miR-204 triple mutants (Figure 3—figure supplement 2). Generating triple mutants was very challenging given the extremely limited genomic targets available for disrupting the microRNAs, but we succeeded in generating animals with CRISPR mutants deleting all or part of the seed sequences at the miR-204-1, miR-204-2, and miR-204-3 loci (Figure 3—figure supplement 2A). Homozygous triple mutant animals lack detectable expression of qPCR-amplifiable mature miR-204 (Figure 3—figure supplement 2C), but they do not exhibit a quantifiably significant loss of thoracic duct formation (Figure 3—figure supplement 2D). The latter result was unexpected, but for reasons detailed below involving a large amount of other newly generated data in this revision, our findings strongly suggest that miR-204 function is indeed regulating lymphatic development and that an unknown compensatory mechanism limits the phenotype in our triple mutants.

Newly generated data in this revision supporting a lymphatic role for miR-204 include:

i) Combined loss of miR-204-1 and miR-204-2 (but not miR-204-1 and miR-204-3) results in identical lymphatic phenotypes regardless of whether miR-204-1 is being knocked down using a morpholino (Figure 2—figure supplement 2) or eliminated using a genetic mutant (Figure 3). Our new findings using the miR204-1 mutant injected with either miR-204-2 or miR-204-3 morpholinos (Figure 3) replicate and fully confirm our previous morpholino-only results (Figure 2—figure supplement 2) showing that miR-204-3 is dispensable.

ii) The loss-of-lymphatic phenotype of miR-204-1^-/-^ mutants injected with miR-204-2 morpholinos is “rescued” by endothelial-specific expression of miR-204-1 (Figure 4E-K). These new data strongly confirm that the lymphatic phenotype of miR-204-deficient animals is due to loss of endothelial miR-204 function, and that miR-204 is required autonomously in the endothelium for generating lymphatics. These data further support our results indicating that the phenotype is not due to nonspecific morpholino effects.

iii) miR-204-deficient animals do not have a defect in either the formation or the initial growth of the secondary sprouts derived from the cardinal vein. Secondary sprouts in these animals fail to halt at the horizontal myoseptum and form the parachordal line as they do in control animals (Figure 2C), but instead continue to grow dorsally and in many cases contribute to veins (Figure 2D; Figure 2—video 1; Figure 3—figure supplement 1). The unusual phenotype of abnormal ongoing dorsal growth of secondary sprouts, together with the otherwise normal overall development of miR-204 deficient animals, suggests that failure to form the parachordal and subsequently thoracic duct is not a result of nonspecific developmental problems in these animals. The abnormal secondary sprout growth and patterning phenotype we have now demonstrated in miR-204 deficient animals is a unique phenotype that we have not observed in the many other lymphatic-defective mutants and morphants we have examined over the years.

We would note again that knockdown with the single pan-204 morpholino targeting all mature miR-204 sequences or knockdown with a combination of two distinct morpholinos blocking maturation of miR204-1 and miR-204-2 (but not miR-204-3) gives the same very unique lymphatic phenotype described in iii) above.

There is precedent in the literature for zebrafish microRNA mutants failing to exhibit the full phenotypes noted in microRNA knockdown studies. Suppression of the evolutionarily conserved endothelial-enriched microRNA mir-126 causes blood vessel defects in mir-126 morpholino-injected animals (Fish et al., 2008) Zou et al., 2011), although lower dose morpholino injections cause mainly lymphatic defects with relatively minor effects on blood vessels (Chen et al., 2016). Interestingly, mir-126 mutant zebrafish also lack the vascular phenotype described in the previous morpholino studies and display only the defects in lymphatic vessel development seen with partial knockdowns (Kontarakis et al., 2018), suggesting some compensation may be taking place in miR-126 mutants.

Reviewer #1:

[…] However, the role of miR-204 in lymphatic vessel formation is not yet conclusive and requires further clarification and investigation. It is still uncertain whether miR-204/NFATC1 interaction contributes to lymphatic vessel formation or patterning. Furthermore, how the expression of miR-204 is regulated during lymphatic vessel development needs to be investigated.1) In Figure 1, how is the expression of miR-204 being regulated during lymphatic vessel development? Is miR-204 always highly expressed in developing lymphatic endothelial cells (i.e. zebrafish lymphatic endothelial cells at 5dpf) and mature lymphatic endothelial cells (i.e. HMVEC-dLy), or is it dynamically regulated in a time-dependent manner?

As noted by the reviewer, we observed high levels of miR-204 expression in both developing zebrafish lymphatics (Figure 1F) and in mature human endothelial cells (Figure 1C), suggesting that miR-204 is maintained at high levels from initial lymphatic development. The mechanisms responsible for establishing the lymphatic vs. blood endothelial expression of miR-204 would certainly be a topic of great interest for future studies.

2) In Figure 2C, what is the authors' opinion on the cause of different phenotypes depending on the dose of morpholinos?

To avoid potential off-target effects from morpholino injections we conducted careful dose response curves to select doses that do not generate obvious gross morphological or other abnormalities. At the 0.5 ng Pan-204 MO dose we detect no noticeable non-vascular abnormalities, while at the higher 0.75 ng dose we see additional effects including smaller eyes, pericardial edema, and craniofacial deformation. While these phenotypes may be associated with reduced miR-204 function, we decided to use the 0.5 ng to avoid potential secondary effects on lymphatics caused by these other abnormalities. This panel was moved to Figure 2—figure supplement 1C.

3) It is still uncertain whether miR-204/NFATC1 molecular pathway contributes to lymphatic vessel formation or patterning. In Figure 2E-N, Figure 3 and Figure 4, authors showed that miR-204 contributes to lymphatic vessel formation by loss-of and gain-of-function experiments. On the other hand, in Figure 6, nfatc1 seems to be required for lymphatic vessel patterning, not lymphatic vessel formation. According to the references the authors provided (Norrmen et al., 2009, Kulkarni et al., 2009), nfatc1 in mice also plays roles in lymphatic vessel patterning, not lymphatic vessel formation. Overall, miR-204 seems to work not only with nfatc1 signaling pathway but also with other signaling pathways, especially in lymphatic vessel formation. The authors need to clarify this issue.

In response to this and other comments we carried out additional experiments to provide more information on how suppressing the miR-204 pathway is interfering with proper lymphatic vascular development. As noted above, we carried out time-lapse imaging of “secondary sprouts,” early prelymphatic sprouts that emerge from the cardinal vein. Our results show that secondary sprouts initially form normally in miR-204 deficient animals and grow normally up to the level of the horizontal myoseptum. However, while secondary sprouts in control animals stop at the myoseptum, turn laterally, and form the parachordal lines (Figure 2C), secondary sprouts in miR-204 deficient animals fail to stop at the level of the myoseptum but continue growing dorsally, in many cases eventually contributing to the venous circulation (Figure 2D, Figure 2—video 1, and Figure 3—figure supplement 1). As also noted above, while nfatc1 axis is not the only gene downstream from miR204, knocking down nfatc1 in the mir-204-deficient animals successfully rescues the lymphatic phenotype. This result suggests that nfatc1 pathway is at least a major important downstream target.

4) In line with comment 3, in Figure 7, it is unclear how deficient 'lymphatic vessel formation' by Pan-204 MO could be rescued by 'lymphatic enlargement' induced by nfatc1 MO. Careful analysis is required.

From our data it appears that increased levels of NFATC1 (in miR-204-deficient animals) lead to failure of secondary sprouts to form parachordals and subsequently lymphatics, while reduced NFATC1 leads to lymphatic enlargement. The relationship between these phenotypes is indeed unclear – they may reflect differential engagement of the same or different downstream pathways. Clearly understanding the molecular pathways and cellular processes activated or suppressed by too little or too much NFATC1 will be a very interesting area for future research investigation, but this is beyond the scope of the present study, which already goes from an unbiased in vitro screen identifying lymphatic microRNAs, to identification of a functionally important microRNA, to the identification and initial functional study of a key target for this microRNA.

Reviewer #2:

[…] Major issues:1) miR-204 has been previously analysed in zebrafish and medaka and prominent expression reported in epithelia and the eye (retinal epithelia and lens). In these previous studies using LNA in situ hybridisation (Conte et al. 2010, Weinholds et al., 2005 paper) there was no indication of vascular expression. The authors should provide evidence with a similar approach that miR-204 is expressed autonomously in intact embryos in the vasculature.It is possible that the function of miR-204 may be non-autonomous in the current report and so at least showing in situ expression in tissues would improve confidence in the current data.

In response to this and other reviewer comments, we have now carried out additional experiments that show that endothelial cell-autonomous expression of mir-204 “rescues” the lymphatic defects in miR-204-1^-/-^ mutants injected with a morpholino blocking maturation of miR-204-2 (Figure 4E-K), and causes precocious thoracic duct development in wild type animals (Figure 4A-D). These new data strongly confirm that miR-204 function is required autonomously in the endothelium for lymphatic development.

We attempted LNA in situ for miR-204 but the sensitivity in our experiments was not sufficient to detect vascular expression, unlike the eye where miR-204 is more highly expressed. LNA in situ hybridization is an excellent method for detecting highly abundant transcripts, but very challenging and not always successful for relatively low copy number transcripts, particularly when dealing with small RNAs where options for design of the probe are extremely limited. However, our qPCR on the FACS-sorted arterial, venous, and lymphatic endothelial cells from transgenic zebrafish shows that miR-204 is highly enriched in the lymphatic endothelial cell population (Figure 1F).

2) The paper relies heavily on the use of MO knockdown which has become increasingly controversial in the zebrafish field. Having one mutant in miR-204-1 is welcome and does improve confidence. However, while this reviewer appreciates that asking for triple mutants is perhaps too much, there are some inconsistencies that are concerning. For example, MO1 + MO2 gives a loss of lymphatics but the mutant for 204-1 + MO2 does not. How can this be explained? Can additional evidence such as transient CRISPR for 204-2 or -3 in the miR-204-1 mutant or similar be provided to further improve confidence in these data with multiple overlapping approaches?

We have carried out a variety of additional experiments for this revision to further substantiate the important role of miR-204 in lymphatic development (as also discussed in detail above), including showing that a mutant for 204-1 + MO2 gives the same loss of lymphatics phenotype as MO1 + MO2:

We now show that combined targeting of miR-204-1 and miR-204-2 gives a lymphatic phenotype but combined targeting of miR-204-1 and miR-204-3 does not, regardless of whether miR-204-1 is targeted using a mutant or a morpholino (Figure 2—figure supplement 2, Figure 3). Although in our original submission we targeted both miR-204-2 and miR-204-3 together with morpholinos in the miR204-1 mutant, we had not actually previously done the experiment noted above by the reviewer of targeting only miR-204-2 with a morpholino in the miR-204-1 mutant.

We now show that transgenic endothelial-specific expression of miR-204 can both rescue the lymphatic defect in mir-204 deficient animals (Figure 4E-K) and cause precocious thoracic duct development in wild type animals (Figure 4A-D).

We now show that miR-204 deficient animals have a striking and unusual phenotype of misdirected secondary sprouts that fail to stop at the horizontal myoseptum to form the pre-lymphatic parachordal line (Figure 2C), but instead continue to grow dorsally and in many cases contribute to veins (Figure 2D; Figure 2—video 1; Figure 3—figure supplement 1).

3). The analysis of phenotype is very superficial. Is specification of LECs impacted? Is LEC cell number at the parachordal line or thoracic duct quantitatively reduced? Is there signalling induced downstream of Vegf-c and Vegf-r3? Eg. pERK such as shown in Shin et al., 2016.

In response to these and other comments, we have carried out a more detailed analysis of the miR204 deficient phenotype, showing that secondary sprouts form and grow to the horizontal myoseptum as in wild type animals, but instead of forming the pre-lymphatic parachordal line at the horizontal myoseptum they continue to grow dorsally and at least some contribute to veins (Figure 2C, D; Figure 2—video 1; Figure 3—figure supplement 1). While a more detailed molecular characterization of the role of miR-204 would be of interest for future studies, we believe it is beyond the scope of the present study, which already goes from an unbiased in vitro screen identifying lymphatic microRNAs to identification of a functionally important microRNA, to the identification and functional validation of a key target for this microRNA.

Along the same lines the phenotypic analysis of the overexpression transgenic for miR-204 and in the NFATC1 vessels should include cell number counts for LECs. This is important as scoring vessel area in 2D images could indicate increased LECs or increased vessel dilation (more luminal content). The NFATC1 mutant could be a fluid imbalance phenotype and unrelated to the miR-204 overexpression phenotype or miR-204 mutant/MO phenotype without further more careful phenotypic analysis.

We have not observed an increase in the number of LEC in the thoracic ducts of nfatc1-deficient animals using counts of endothelial cell nuclei. The reason for the lymphatic hyperplasia phenotype (also noted in mice) remains unclear at this point, but we have not observed defects in flow through the thoracic duct in preliminary lymphatic drainage experiments (data not shown).

4). The overexpression of miR-204 gives premature thoracic duct development but the embryo shown looks older than the control (increased distance between DA and PCV). The experiment is also under-controlled. Please provide analysis of markers of other tissues to control for staging differences (eg. rag1 expression in thymus is a useful marker that comes on progressively from around 2.5 dpf). Please also show how much the endothelial levels of miR-204 are increased in the transgenic – miRNA's are typically highly expressed and Figure one suggests very high levels of miR-204 already in LECs, so why would one expect such a phenotype upon over-expression?

We carefully repeated the miR-204 germline transgenic overexpression experiment together with measurement of the distance between DA and PCV, and observed precocious thoracic duct formation without change in the DA/PCV distance (Figure 4—figure supplement 1). We measured an approximately 2-fold increase in miR-204 levels in these germline transgenic animals (Figure 4—figure supplement 1E).

To address this question experimentally in another and perhaps more direct way, we also showed that mosaic endothelial-specific expression of miR-204 in otherwise wild type, normally-developing animals also leads to precocious thoracic duct formation (Figure 4A-D).

5) Is NFATC1 expressed in LECs and the PCV in zebrafish by ISH? Is the expression increased upon progressive loss of miR-204 in the mutants and MO scenarios?

Previous reports have noted nfatc1 expression in the endothelium in various species including zebrafish (e.g. Coxam et al., Cell Reports 7:623, 2014), and we and others have shown that NFATC1 is differentially expressed at higher levels in LEC. We show that miR-204 mimic decreases and miR-204 inhibitor increases NFATC1 expression in HUVEC in vitro(Figure 5C) and that miR-204 deficient zebrafish have increasedin vivonfatc1 expression (Figure 5F), clearly demonstrate that NFATC1 is a target of miR-204 regulation.

6) The analysis of NFATC1 phenotype to correlate with miR-204 phenotypes relies exclusively upon MO knockdown and the use of CsA, an inhibitor commonly considered to have broad impact on embryos. Unlike the miR targeting, where it is difficult to expect generation of triple mutants, analysis of a stable NFATC1 mutant does not seem an unreasonable thing to ask for. It is likely that a mutant may be available already. The authors should provide more confidence in their increased lymphangiogenesis phenotype upon NFATC1 loss of function by including the analysis of a mutant strain. Alternatively (or additionally), they could consider targeting the miR-binding site in the 3'UTR to definitively demonstrate their mechanism.

In response to this and other reviewer comments, we now demonstrate that two separate nfatc1 morpholinos, cyclosporine treatment targeting NFAT signaling, and a newly generated nfatc1 mutant all result in the same lymphatic enlargement phenotype (Figure 6, Figure 6—figure supplement 1).

7) The honing in on NFATC1 as a target came without explanation of how many miR-204 targets are predicted in vasculature. Providing additional bioinformatic prediction would improve the study. How many LEC transcripts have predicted miR-204 target sites? Is this statistically enriched over other cell types? How does NFATC1 rank as a predicted target taking into account the number of predicted sites, homology etc?

We provide a detailed response to this comment in the “Explain our focus on NFATC as a key miR-204 target” section (under “New experimental data provided for revision”) above. As we noted previously, we recognize of course that microRNAs can have hundreds of potential targets, and our original goal in testing nfatc1 had been just to validate that miR-204 did indeed target and regulate at least one gene known to be important for lymphatic development. It was a bit of a surprise to us that nfatc1 knockdown or mutants gave us a nice lymphatic phenotype, and a big surprise that we could so effectively rescue miR-204 deficiency by knocking down nfatc1.

Reviewer #3:

[…] This study is well organized and written in a logical manner. There are still several points that should be addressed to support their claim.1) It is unclear why the authors need to compare zebrafish and human? The conservation of mir-204-dependent regulation of NFATC1 among zebrafish and human might be interesting: however, the authors could identify mir-201 using mir-RNAs from the Tg(mrc1a:eGFP)^y251^ and Tg(kdrl:mCherry)^y171^ as shown in Figure 1F. Because the importance of mir-204 in zebrafish is focused in this study, the list of mir using the mir from the FACS-sorted cells should be used as an initial material. The authors might do this experiment easily using their Tg fish.The result shown in Figure 1A, B, C should become Figure supplement.

We used both human and zebrafish models for several reasons, but mainly (i) to ensure that we were looking at important evolutionarily conserved microRNA regulators of lymphatic development, and (ii) to provide complementary in vitroand in vivo models enabling a full range of experimental paradigms to be employed. We began by carrying out the small RNAseq using human cells mainly for technical reasons (eg, obtaining ample starting material for preparation of highly representative and replicated sequencing libraries), but also to permit comparison to previous screens carried out using human cells.

2) In addition to #1, time course of mir expression should be demonstrated at least at two points when the secondary sprout starts and when thoracic duct formation is completed. The authors merely showed the expression of mir-204 in the zebrafish embryos at 5 dpf.

It is not technically possible to obtain sorted lymphatic/lymphatic progenitor cells from our double transgenic animals at early stages when secondary sprouting begins – there are extremely few lymphatic progenitor cells at this stage to begin with and these cells are not yet “single labeled” with only GFP, so they cannot be sorted out as a separate population.

3) The requirement of mir-204 is partially proven in this study. As the authors explained in Figure 3, they need to do the similar experiments using mir-204-2 ^-/-^ + MO1, 3 and mir-204-3 ^-/-^ + MO1, 2. It is not so difficult to complete these experiments by establishing mir-204-2 mutant and mir-204-3 mutant. They describe the detail of mir-204-2 and mir-204-3 in Figure 2—figure supplement 1.

As noted in the “Analyze mir-204 mutants” section (under “New experimental data provided for revision”) above, we have added a large amount of new data reinforcing our conclusion that miR-204 is required cell-autonomously for proper lymphatic development.

4) The potential target of mir-204 is not described well. The authors described this in the Discussion. The reason why Nfatc1 was picked up is not clearly described in the present manuscript. The authors examined Nfatc1 following the prediction of RNA22 database. There are several microRNA target databases beside RNA22. Thus, targets expected in other database should be examined as a negative control.

Please see the “Explain our focus on NFATC as a key miR-204 target” section (under “New experimental data provided for revision”) above, where we provide a more complete explanation of how and why we selected NFATC1 for further analysis. Again, our original intent was merely to demonstrate that miR-204 modulated a factor previously shown to be important for lymphatic development. The nfatc1 morphant/mutant phenotypes and especially the successful rescue of miR-204 deficient animals by inhibiting nfatc1 expression were welcome surprises!

5) To more precisely show the hierarchical regulation of mir-204 and nfatc1, the authors can examine the expression of nfatc1 using FACS-sorted LECs of Tg8mrc1a:eGFP)^y251^ total morphants (mir-204-1, -2,-3 morphants).

We report that miR-204 deficient zebrafish have increasedin vivonfatc1 expression (Figure 5F). We also show that miR-204 mimic decreases and miR-204 inhibitor increases NFATC1 expression in HUVEC in vitro(Figure 5C). These results demonstrate that NFATC1 is a target of miR-204 regulation.

6) It is unclear whether secondary sprout is affected in mir-204 deficient embryos. Although there are no parachordal lines in the scheme of Figure 7L, there is no evidence of impairment of secondary sprout followed by parachordal line development in the present study. Although the authors focused on thoracic duct formation, prior to this, secondary sprout and parachordal lines could be analyzed. No description in Figure 3.

In response to this and other comments from our reviewers, and as described in detail in the “Examine early stages of lymphatic development in miR-204-deficient animals” section (under “New experimental data provided for revision”) above, we now include new time-lapse imaging data showing that secondary sprouts in miR-204 deficient animals emerge and grow to the level of the horizontal myoseptum as in normal animals but then fail to form the pre-lymphatic parachordal lines and instead continue to grow dorsally, in at least some cases contributing to veins (Figure 2C, D; Figure 2—video 1; and Figure 3—figure supplement 1).

7) The authors mentioned "lymphatic specification" in Discussion. They can interpret their data when carefully examine the effect of depletion of mir-204 and nfatc1 on secondary sprout and subsequent parachordal lines.

Please see our comments in the response to point #6 above, and in the “Examine early stages of lymphatic development in miR-204-deficient animals” section (under “New experimental data provided for revision”). Our new data shows that secondary sprouts form and grow normally to the myoseptum but are then misdirected and fail to contribute to the parachordal lines. In addition to the new data we have added new text to our manuscript discussing these findings.